# MetaphorVU: Towards Metaphorical Video Understanding

**Zhuoqun Li** [1 2]   **Boxi Cao** [1]   **Guiping Jiang** [3]   **Fangrui Lv** [4]   **Ruotong Pan** [3]   **Jianan Wang** [3]   **Xiangyu Wu** [3]
**Hongyu Lin** [1]   **Yaojie Lu** [1]   **Yong Du** [3]   **Ruyin Jia** [3]   **Liyan** [3]   **Tingting Gao** [3]   **Han Li** [3]   **Xianpei Han** [1]   **Le Sun** [1]

## Abstract

Metaphorical videos are prevalent across various real-world scenarios to convey complex ideas, and understanding them typically requires high-order cognitive capabilities. The lack of systematic studies on metaphorical video understanding not only constrains the real-world applicability of MLLMs but also impedes the thorough assessment of their high-order cognitive capabilities. To bridge this gap, we propose MetaphorVU-Bench, the first systematic and comprehensive benchmark dedicated to metaphorical video understanding. Through experiments, we find current MLLMs struggle with accurate metaphorical video understanding, lagging far behind human level, primarily due to defective cross-domain mapping. Motivated by this finding, we construct a metaphor knowledge graph as mapping augmentation and propose MetaphorBoost, an inference-time enhancement framework achieving consistent performance improvement. Our benchmark, analysis, and method provide useful insights and a foundation for future research on advancing MLLMs. Code: https://github.com/icip-cas/MetaphorVU.

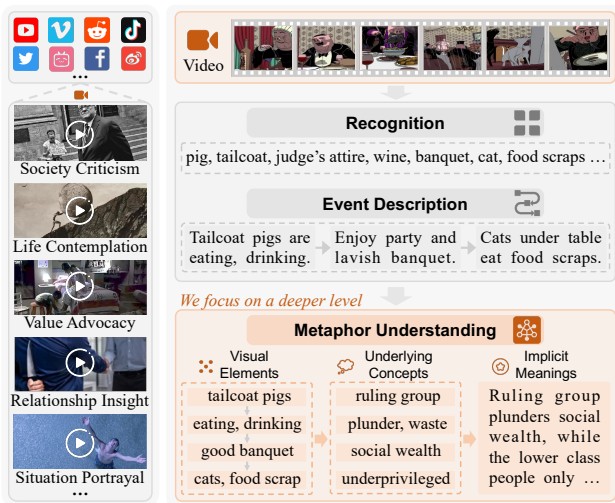

*Figure 1.* Metaphorical videos are prevalent across various real-world scenarios to convey many complex ideas, and metaphorical video understanding requires high-order cognitive capabilities.

## 1. Introduction

Metaphorical videos serve as a crucial medium for conveying complex ideas in human society, and they widely exist in important scenarios such as social media and public communication (Krippendorff, 1993; Shifman, 2013; Burgers et al., 2016; Shutsko, 2020). Rather than directly presenting profound meanings such as society criticism and life contemplation, video creators often employ metaphorical

content to guide viewers toward associations and interpretations (Johnson & Malgady, 1979; Camac & Glucksberg, 1984; Zhang, 2021; Alnajjar et al., 2022). According to multimodal metaphor theory, human understanding of metaphorical videos is a high-order cognitive process that transforms perceived signals into deeper semantics, with the core lying in cross-domain mapping that links visual elements to underlying concepts (Forceville et al., 2009; Fahlenbrach, 2016; Pan & Tay, 2020; Zhang, 2021). As illustrated in Figure 1, humans can link visual elements (e.g., *tailcoat pigs, banquet, and cats under table*) with underlying concepts (e.g., *ruling group, social wealth, and underprivileged*), thereby revealing implicit meanings of *critique toward the ruling group and sympathy for the lower class people*.

Recently, multimodal large language models (MLLMs) have been widely used in practical applications and significantly pushed the frontier of video understanding capabilities (OpenAI, 2025; Bai et al., 2025a; An et al., 2025; Google, 2025b). Unfortunately, most existing work focuses on literal perception tasks such as object recognition and event description of videos (Li et al., 2025c; Bandraupalli et al., 2025; Brkic et al., 2025; Liu et al., 2025), lacking a systematic study of high-order cognitive metaphorical video understanding.

---
[1]Chinese Information Processing Laboratory, Institute of Software, Chinese Academy of Sciences [2]University of Chinese Academy of Sciences [3]Kuaishou Technology [4]Department of Automation, Tsinghua University. Correspondence to: Boxi Cao <caoboxi@iscas.ac.cn>, Xiangyu Wu <wuxiangyu06@kuaishou.com>.

*Proceedings of the 43rd International Conference on Machine Learning*, Seoul, South Korea. PMLR 306, 2026. Copyright 2026 by the author(s).

This gap makes it difficult to assess whether MLLMs can accurately transform perceived visual signals into deeper semantics like humans, limiting their reliable application in many complex scenarios and further improvement of cognitive capabilities (Shutsko, 2020; Zhang, 2021; Alnajjar et al., 2022; Okonski et al., 2022). Therefore, effectively evaluating and advancing the metaphorical video understanding capability of MLLMs is of great significance for their widespread utilization and further enhancement.

To this end, we propose **MetaphorVU-Bench**[1], the first comprehensive benchmark for metaphorical video understanding, characterized by a well-founded systematic taxonomy, metaphorical videos curated from billions of real-world candidates, and rigorous human annotation. Specially, to ensure a systematic evaluation, as illustrated in Figure 2, we first design a well-founded video metaphor taxonomy, covering 8 types of video metaphor grounded in multimodal metaphor theory (Forceville et al., 2009; Forceville & Urios-Aparisi, 2009) and its extensions (Bordwell, 2013b; Stam, 2017; Schechner, 2017; Chandler, 2022). Guided by this taxonomy, as illustrated in Figure 3, we construct the benchmark sourced from the real world with careful filtration and rigorous annotation. Firstly, to ensure the evaluation accurately reflects practical performance, we source data from a real-world video platform covering diverse topics. Secondly, to efficiently select metaphorical videos from billions of sources, we apply a multi-stage filtration based on video information and comments, yielding 860 videos spanning the taxonomy. Finally, to obtain reliable metaphor interpretations, we conduct manual annotation with strict cross-validation, yielding a high-quality benchmark for systematic evaluation of metaphorical video understanding.

Based on above MetaphorVU-Bench, we systematically evaluate 11 representative close-source and open-source MLLMs. Experimental results show that current MLLMs still struggle with accurate metaphorical video understanding. Even the most advanced MLLMs, such as Gemini-3-Pro and GPT-5, can only achieve average scores around 64, significantly lagging behind human-level performance by nearly 20 points. Furthermore, to better understand causes of MLLM failures and develop targeted optimization methods, we conduct an error analysis across MLLMs of varying capabilities. Analysis results reveal that over 80% of failures do not stem from recognition error, but rather from defective cross-domain mapping, where current MLLMs fail to effectively establish links from visual elements to underlying concepts. These findings indicate that enhancing cross-domain mapping is the key to improving MLLMs performance on metaphorical video understanding.

Motivated by above findings, rather than relying on MLLMs

to perform blind cross-domain mapping, we propose a novel enhancing framework, **MetaphorBoost**, utilizing a metaphorical knowledge graph as external cognitive scaffold to augment cross-domain mapping. Specifically, to provide MLLMs with metaphor-specific interconnected augmentation, we construct the first metaphorical knowledge graph by collecting metaphorical texts, extracting metaphorical concepts and connecting these concepts. At inference time, MetaphorBoost queries the metaphorical knowledge graph based on content recognition results to obtain reliable references, thereby promoting cross-domain mapping and precise metaphor interpretations. Experimental results show MetaphorBoost achieves consistent performance improvements across multiple MLLMs, providing a preliminary exploration and foundation for future research. Main contributions of this paper can be summarized as follows:

- We propose MetaphorVU-Bench, which is the first benchmark dedicated to systematic and comprehensive evaluation for metaphorical video understanding.
- We conduct extensive experiments and analysis, revealing the deficiencies of current MLLMs and providing insights into the underlying causes of their failures.
- We construct MetaphorBoost, boosting metaphorical video understanding via inference-time mapping augmentation based on a metaphorical knowledge graph.

## 2. MetaphorVU-Bench

The lack of systematic research on metaphorical video understanding to some extent limits further application reliability and capability enhancement of MLLMs. To bridge this gap, we design the first systematic video metaphor taxonomy and construct MetaphorVU-Bench based on this taxonomy, enabling systematic evaluation of metaphorical video understanding. In this section, we sequentially present the taxonomy, benchmark and evaluation method.

### 2.1. Video Metaphor Taxonomy

To ensure reliable and principled evaluation of metaphorical video understanding, a systematic video metaphor taxonomy is essential for building the benchmark. Therefore, we draw on multimodal metaphor theory (Forceville et al., 2009; Forceville & Urios-Aparisi, 2009) and its extensions in the video field (Bordwell, 2013b; Stam, 2017; Schechner, 2017; Chandler, 2022), designing the first systematic video metaphor taxonomy. Specifically, as illustrated in Figure 2, video metaphor can be categorized as following 8 types:

- *Body Language.* Video conveys implicit meanings through body movements of characters, typically some exaggerated or semantically meaningful actions.
- *Atmosphere Language.* Video conveys implicit meanings by environmental atmosphere, such as purposeful

---

[1]The proposed benchmark of this paper is released in https://huggingface.co/datasets/lzq2021/MetaphorVU-Bench.

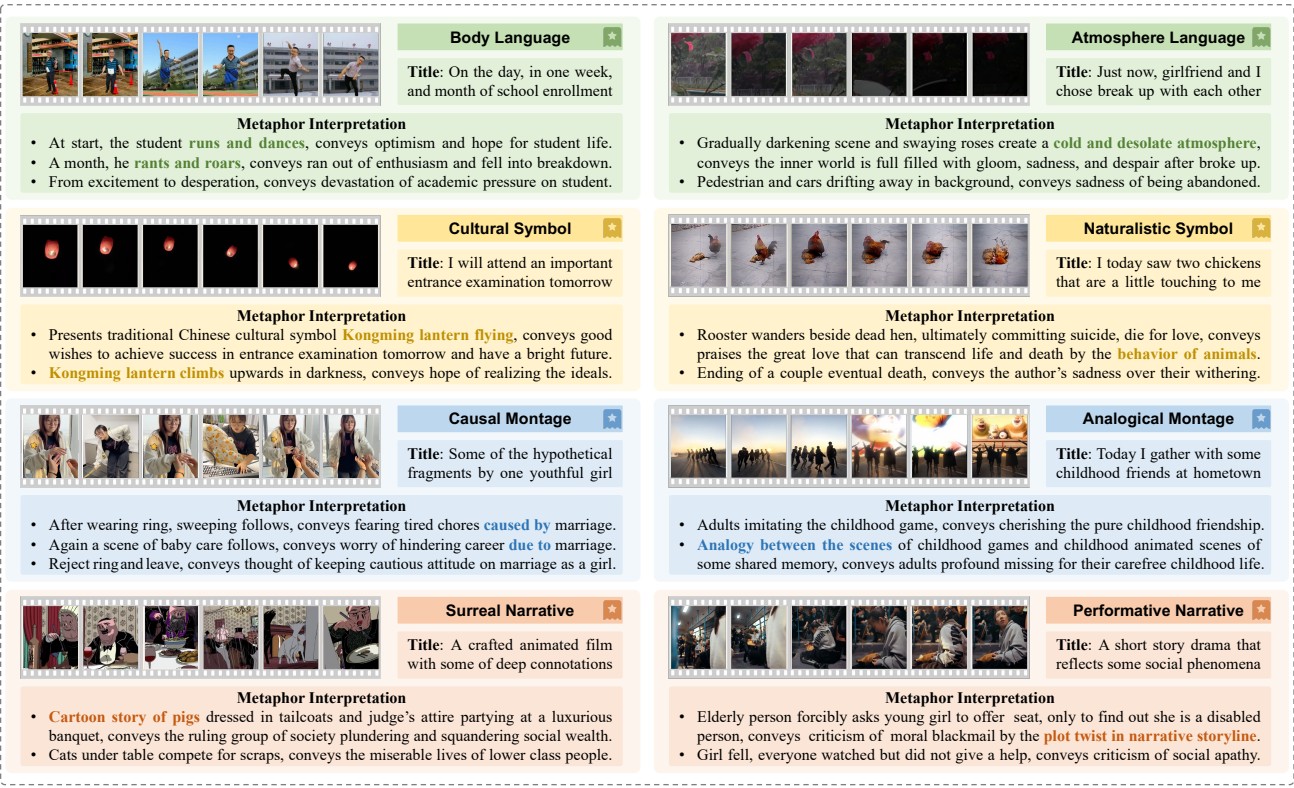

*Figure 2.* MetaphorVU-Bench contains 8 types of video metaphor, enabling systematic evaluation of metaphorical video understanding. Note that most videos simultaneously contain multiple types of metaphor, we only show the dominant one in each case for illustration.

variations in the color, lighting and composition.

- *Cultural Symbol.* Video conveys implicit meanings by symbolism of cultural artifacts, such as flying China Kongming lanterns or building a Christianity cross.

- *Naturalistic Symbol.* Video conveys implicit meanings by symbolism of natural elements, such as animal behaviors, plant growth, and changing starry skies.

- *Causal Montage.* Video conveys implicit meanings through juxtaposing cause-and-effect shots to guide audiences to infer some causal logic in their brain.

- *Analogical Montage.* Video conveys implicit meanings by juxtaposing visually or thematically similar shots to guide audiences to infer analogical logic in brain.

- *Surreal Narrative.* Video conveys implicit meanings through characters and plots transcending physical constraints, such as cartoons and AI-generated videos.

- *Performative Narrative.* Video conveys implicit meanings through dramatized storytelling performed by human actors, such as short play in video platforms.

This video metaphor taxonomy provides a solid foundation for building a comprehensive benchmark and conducting systematic evaluation. Examples for each type are illustrated in Figure 2. Detailed theoretical basis for the taxonomy is shown in Appendix A, more examples are in Appendix H.

## 2.2. Benchmark Construction

Based on above video metaphor taxonomy, we construct MetaphorVU-Bench, enabling systematic evaluation of metaphorical video understanding. Specifically, as shown in Figure 3, we select real-world data source, apply efficient multi-stage filtration and perform reliable manual annotation, obtaining the benchmark with strict quality validation. This benchmark encompasses diverse video topics, with sufficient data volume and suitable video duration for evaluation. Thematic diversity is shown in Figure 4. Statistics of sample number, video duration and token number of golden interpretation are shown in Table 1. In the following, we provide detailed process of benchmark construction.

**Real-world Data Source.** We prioritize diversity and authenticity when selecting data source, which are two critical factors for credible evaluation. Specially, to ensure evaluation results can accurately reflect metaphorical video understanding capability in real world, the benchmark should cover diverse video topics from daily life. Moreover, since current MLLMs mainly support inputting a limited number of frames, the benchmark should contain videos with compatible durations to avoid video length becoming a confounding factor. Therefore, we use Kaishou[2] short-video

---

[2]https://www.kuaishou.com/?isHome=1

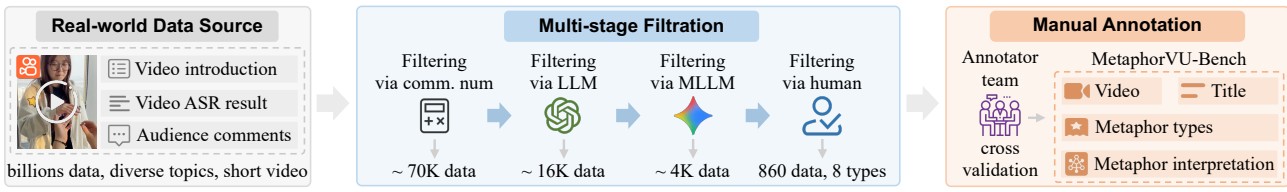

*Figure 3.* We construct MetaphorVU-Bench by using a real-world short-video platform as source, selecting metaphorical videos from a large-scale video pool through multi-stage filtration, and manually annotating video metaphor interpretations with rigorous quality control. MetaphorVU-Bench can effectively support systematic and comprehensive evaluation of metaphorical video understanding.

*Table 1.* Benchmark statistics of sample number, average video duration and average token number of golden interpretations.

| Type | # Samples | Avg. Duration (s) | Avg. Tokens |
|------|-----------|-------------------|-------------|
| Body Language (Body L.) | 136 | 32.2 | 111.3 |
| Atmosphere Language (Atmosp. L.) | 150 | 13.1 | 104.5 |
| Cultural Symbol (Cultural S.) | 62 | 23.5 | 114.4 |
| Naturalistic Symbol (Natural. S.) | 113 | 17.3 | 108.8 |
| Causal Montage (Causal M.) | 54 | 57.7 | 108.9 |
| Analogical Montage (Analog. M.) | 171 | 58.7 | 124.8 |
| Surreal Narrative (Surreal N.) | 112 | 30.4 | 117.1 |
| Performative Narrative (Perform. N.) | 62 | 86.8 | 118.6 |
| MetaphorVU-Bench | 860 | 37.2 | 114.2 |

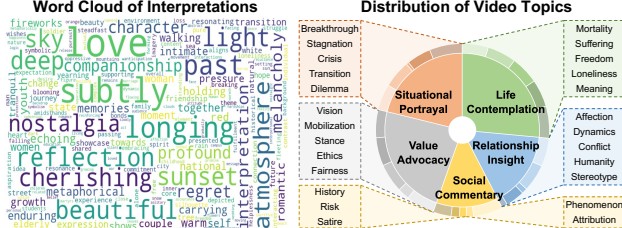

*Figure 4.* Benchmark covers diverse video topics, enabling accurate evaluation of real-world metaphorical video understanding.

platform as the data source, which can provide massive real-world videos spanning a wide range of topics and video duration is compatible with most common-used MLLMs.

**Efficient Multi-stage Filtration.** The data source contains billions of videos, of which only a small fraction involve metaphorical logic. To efficiently isolate metaphorical videos, we design a multi-stage filtration strategy.

Considering audience comments often contain interpretation of videos, which can serve as an important indicator, we first filter videos by amount of audience comments, retaining only those with more than 150 comments, yielding 70K videos. Then, we use a powerful LLM (GPT-5) to analyze the video introduction, automatic speech recognition (ASR) result and audience comments to determine whether each video contains metaphorical logic, reducing the amount of candidate video set to 16K. The detailed prompt guideline for LLM to do filtration is shown in Appendix B.1.

Furthermore, considering above filtration process does not directly use visual information and LLM analysis may not align with the actual video, we conduct further check and filtration. A powerful MLLM (Gemini-3-Pro) is used to verify whether above analysis is consistent with original videos, reducing the amount of candidate video set to 4K. Then, a human team performs final filtration based on original video, video introduction and audience comments, resulting in 860 videos with definite metaphorical logic. Additionally, annotators identify the metaphor type for each video, balancing the number of samples across each metaphor type as much as possible. The prompt for MLLM and human annotators filtration are in the Appendix B.2 and B.3, respectively.

**Reliable Manual Annotation.** Since video metaphor interpretation is a flexible text, different annotators may produce varying linguistic styles and formats. Although these interpretations may all be substantively correct, such subjectivity and format inconsistency make it difficult to conduct evaluation by the benchmark. Therefore, when annotating video metaphor interpretation, we require human annotators to reference video introduction and audience comments and follow a fixed format (i.e., *specifying which visual elements convey which implicit meanings*). This can reduce subjectivity and enhance format consistency, thereby improving the reliability of benchmark. Additionally, annotators are responsible for providing a brief title that introduces necessary background information of the video. The guideline for manual annotation is shown in Appendix B.4.

**Strict Quality Control.** To further ensure benchmark quality, we employ cross-validation among annotators to avoid errors by individual oversight. During the final video filtration stage, we assign three annotators for each candidate video. If any annotator considers the video to lack definite metaphorical logic, the video is excluded. During the interpretation annotation stage, we assign one interpreter and two reviewers for each video. The initial annotation from interpreter is reviewed by reviewers, and all three iteratively refine it until reaching a good metaphor interpretation that is acceptable to all. In additional, to avoid speech and subtitles in videos directly unveiling the metaphorical meanings, we apply muting and subtitle removal using open-source tool[3] before manual annotation, ensuring both annotation

---

[3] https://github.com/YaoFANGUK/video-subtitle-remover

*Table 2.* Overall results on MetaphorVU-Bench. To intuitively demonstrate gap between MLLMs and human*, we sample 100 instances and collect human-written metaphor interpretations as upper-bound. The table shows that current MLLMs exhibit limited capability, and existing reasoning-enhanced methods fail to achieve effective improvements. In contrast, our method proves to be more effective.

| Method | Body L. | Atmosph. L. | Cultural S. | Natural. S. | Causal M. | Analog. M. | Surreal N. | Perform. N. | Average |
|---|---|---|---|---|---|---|---|---|---|
| Upper-bound | | | | | | | | | |
| Human* | 87.8 | 87.5 | 89.1 | 83.8 | 72.0 | 81.5 | 78.1 | 78.0 | 83.4 |
| Close-source MLLMs | | | | | | | | | |
| GPT-5 (OpenAI, 2025) | 69.9 | **76.3** | 77.4 | 66.6 | 45.0 | 55.4 | 54.9 | 46.1 | 63.7 |
| GPT-4o (OpenAI, 2024) | 63.4 | 70.5 | 70.3 | 62.6 | 39.1 | 48.2 | 45.7 | 37.9 | 56.8 |
| Qwen3-VL-Plus (Bai et al., 2025a) | 66.8 | 72.5 | 74.8 | 65.5 | 51.5 | 54.2 | 50.4 | 43.7 | 61.4 |
| Gemini-2.5-Pro (Google, 2025a) | 65.5 | 71.3 | 74.3 | 64.4 | 53.5 | 55.7 | 52.1 | 46.9 | 61.8 |
| Gemini-3-Pro (Google, 2025b) | 71.2 | 74.0 | 75.1 | **66.9** | 49.4 | 58.9 | 51.1 | 48.1 | 63.8 |
| Doubao-1.5-Vision-Pro (Guo et al., 2025) | 58.2 | 64.1 | 65.5 | 58.9 | 27.8 | 42.5 | 39.8 | 26.6 | 50.5 |
| Open-source MLLMs | | | | | | | | | |
| Qwen2.5-VL-7B-Instruct (Bai et al., 2025b) | 36.0 | 49.9 | 46.1 | 42.1 | 12.4 | 23.5 | 28.6 | 16.1 | 33.8 |
| Qwen3-VL-8B-Thinking (Bai et al., 2025a) | 56.0 | 66.1 | 68.8 | 60.8 | 33.2 | 45.0 | 39.3 | 29.2 | 52.0 |
| LLaVA-onevision-1.5-8B-Instruct (An et al., 2025) | 35.7 | 47.2 | 47.3 | 45.0 | 13.8 | 21.3 | 27.0 | 21.2 | 38.1 |
| GLM-4.5V (Team et al., 2025) | 62.7 | 67.9 | 71.9 | 62.1 | 37.6 | 50.1 | 46.1 | 38.4 | 56.8 |
| Qwen3-VL-235B-A22B-Thinking (Bai et al., 2025a) | 65.4 | 70.4 | 71.9 | 58.1 | 43.2 | 54.6 | 46.1 | 38.1 | 58.6 |
| Reasoning-enhanced Methods | | | | | | | | | |
| VideoRFT (Wang et al., 2025b) | 38.9 | 52.8 | 48.4 | 46.0 | 13.5 | 24.8 | 27.2 | 16.6 | 35.6 |
| Vision-R1 (Huang et al., 2025) | 39.3 | 45.1 | 42.0 | 42.4 | 19.4 | 23.2 | 25.0 | 18.6 | 33.1 |
| ReAd-R (Long et al., 2025) | 42.1 | 54.1 | 48.9 | 46.3 | 15.7 | 26.4 | 26.2 | 17.6 | 36.8 |
| LTR (Liao et al., 2025) | 54.1 | 44.7 | 56.2 | 47.4 | 27.8 | 44.6 | 31.9 | 36.1 | 44.5 |
| ViTCoT (Zhang et al., 2025a) | 58.8 | 47.7 | 59.2 | 48.7 | 26.1 | 45.1 | 34.0 | 32.1 | 46.2 |
| Prompt Engineering (Wei et al., 2022) | 57.8 | 66.3 | 67.9 | 59.2 | 36.1 | 42.7 | 41.6 | 32.6 | 52.4 |
| Few-shot Example (Dong et al., 2024) | 57.6 | 69.4 | 69.2 | 58.7 | 33.5 | 44.9 | 43.5 | 32.6 | 53.6 |
| Mapping Augmentation via Metaphorical Knowledge Graph | | | | | | | | | |
| MetaphorBoost (Gemini-3-Pro) (Ours) | **71.5** | **76.3** | **77.5** | **66.9** | **57.2** | **59.1** | **57.3** | **50.8** | **66.1** |
| Δ (vs Gemini-3-Pro) | +0.3 | +2.3 | +2.4 | +0.0 | +7.8 | +0.2 | +6.2 | +2.8 | +2.3 |
| MetaphorBoost (Qwen2.5-VL-7B-Instruct) (Ours) | 40.7 | 55.7 | 51.2 | 49.0 | 12.5 | 26.1 | 31.4 | 19.2 | 37.9 |
| Δ (vs Qwen2.5-VL-7B-Instruct) | +4.6 | +5.8 | +5.1 | +6.9 | +0.1 | +2.6 | +2.9 | +3.0 | +4.1 |
| MetaphorBoost (Qwen3-VL-8B-Thinking) (Ours) | 61.8 | 71.0 | 71.8 | 61.3 | 36.7 | 47.1 | 45.7 | 31.5 | 55.9 |
| Δ (vs Qwen3-VL-8B-Thinking) | +5.8 | +4.9 | +3.0 | +0.5 | +3.5 | +2.1 | +6.4 | +2.3 | +3.8 |

and evaluation rely solely on visual information of videos.

### 2.3. Evaluation Task and Metric

**Task Formulating.** Based on this benchmark, we evaluate the metaphorical video understanding as following formula:

$$\hat{\tau}, \hat{o} = \mathcal{F}(v \oplus t) \qquad (1)$$

where $\mathcal{F}$ is evaluated system, $v$ is video, $t$ is title, $\oplus$ denotes input combination, $\hat{\tau}$ is thinking process and $\hat{o}$ is output video metaphor interpretation. Generally, MLLMs first recognize visual elements, establish linking to underlying concepts and reveal implicit meanings in $\hat{\tau}$, then formally interpret which visual elements convey which implicit meanings in $\hat{o}$. Detailed evaluation prompt is shown in Appendix C.1.

**Evaluation Metric.** Since video metaphor interpretation is free-form text, rule-based metrics are difficult to provide reliable scores (Mayfield et al., 2024; Li et al., 2025d). Therefore, we follow the metrics in previous free-form video-QA works (Yu et al., 2025; Long et al., 2025), using DeepSeek-V3.2[4] as LLM judge. Specifically, we design detailed scoring guidelines for LLM judge to accurately assess MLLMs output. With golden interpretation as reference, the judge evaluates output interpretation on its accuracy in grounding

metaphorical visual elements and revealing implicit meanings, assigning a integer score from 0 to 10, then rescaled to 0-100 for presentation. Guidelines for LLM judge are in Appendix C.2. Consistency analysis between LLM judge and human judge is in Appendix C.3, where Pearson correlation coefficient is 0.85, confirming the LLM judge is reliable.

## 3. MetaphorVU Evaluation

### 3.1. Evaluation Settings

**Selected Baselines.** To comprehensively evaluate the ability on metaphorical video understanding, we extensively select both close-source and open-source models of various scales, as well as representative reasoning-enhanced methods. Specially, (1) **Close-source MLLMs**, including GPT-5 (OpenAI, 2025), GPT-4o (OpenAI, 2024), Qwen3-VL-Plus (Bai et al., 2025a), Gimini-2.5-Pro (Google, 2025a), Gimini-3-Pro (Google, 2025b) and Doubao-1.5-Vision-Pro (Guo et al., 2025). (2) **Open-source MLLMs**, including Qwen2.5-VL-7B-Instruct (Bai et al., 2025b), Qwen3-VL-8B-Thinking (Bai et al., 2025a), LLaVA-onevision-1.5-8B (An et al., 2025), GLM-4.5V (Team et al., 2025), and the Qwen3-VL-235B-A22B-Thinking (Bai et al., 2025a). (3) **Reasoning-enhanced Methods**, which enhance the reasoning ability of base model by post-training or inference-

---

[4]https://api-docs.deepseek.com/news/news251201

time scaling, including VideoRFT (Wang et al., 2025b), Vision-R1 (Huang et al., 2025), ReAd-R (Long et al., 2025), LTR (Liao et al., 2025), ViTCoT (Zhang et al., 2025a), the first 3 methods are post-training based on Qwen2.5-VL-Instruct, and the last 2 methods are inference-time scaling based on Qwen3-VL-8B-Thinking. Additionally, we add two commonly used inference-time scaling methods based on Qwen3-VL-8B-Thinking, including Prompt Engineering (Wei et al., 2022) with a prompt tailored for metaphorical video understanding, and Few-shot Example (Dong et al., 2024) with 3-shot examples tailored for metaphorical video understanding. More details of baselines are in Appendix F.

**Implementation Details.** To ensure evaluation reliability, we conduct experiments following the general practices. For close-source MLLMs, we directly use official APIs for experiments. For open-sourced MLLMs, we download the weights of models from official repositories and deploy them as APIs using vLLM[5]. For reasoning-enhanced methods, we use officially provided post-training weights or the inference-time scaling strategies specified in their original papers. To ensure consistency, the generation temperature is uniformly set to 0.7 for all models. Regarding the input, since not all MLLMs support direct video input, we follow the common practice by splitting videos into frames and converting them to base64 encoding (Bai et al., 2025b;a), thereby supporting all MLLMs involved in this experiment.

## 3.2. Overall Results

Experimental results of MLLMs and reasoning-enhanced methods are in the Table 2, there are two main conclusions:

**Current MLLMs struggle with accurate metaphorical video understanding.** For open-source MLLMs, table shows there is a significant gap with human, for example, Qwen3-VL-8B-Thinking achieves average score of 52.0, far below the human score of 83.4. For close-source MLLMs, they can generally achieve relatively higher performance, especially Gemini-3-Pro, demonstrating the strongest overall performance among all baselines, with average score of 63.8. However, this performance still falls short of the human level, indicating substantial room for improvement.

**Previous inference-time scaling methods for recognition and event description yield marginal improvement.** LTR and ViTCoT, which are two inference-time scaling methods designed for enhancing object recognition and event description, even degrade performance of base model Qwen3-VL-8B-Thinking. In comparison, our implemented prompt engineering and few-shot examples methods designed for metaphorical understanding yield relatively limited improvements. Furthermore, despite additional data and training overhead, post-training via long chain-of-thought reinforce-

---

[5]https://pypi.org/project/vllm/

*Table 3.* Proportion of each deficiency type, reveals that enhancing cross-domain mapping is key to improving performance.

| Model | Wrong Recognition | Missing Mapping | Superficial Mapping | Improper Mapping |
|---|---|---|---|---|
| Gemini-3-Pro | 10.7% | 27.9% | 33.7% | 27.7% |
| Qwen3-VL-8B-Thinking | 13.5% | 28.1% | 28.3% | 30.1% |

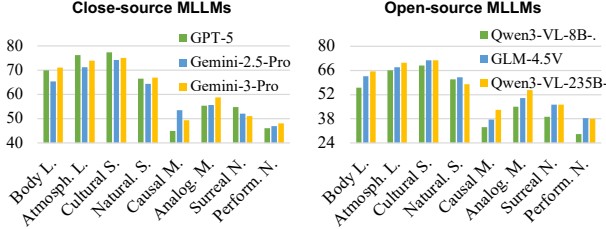

*Figure 5.* Performing worse on subsets requiring more cross-domain mapping, supports importance of mapping augmentation.

ment learning optimized for recognition and description, such as VideoRFT and Vision-R1, only achieve marginal improvements over base model Qwen2.5-VL-Instruct.

## 3.3. Detailed Analysis

**Error Analysis.** To investigate the core deficiencies of MLLMs in detail, we manually observe and identify 4 common types of deficiency in MLLMs thinking process: (1) wrong recognition of visual elements, (2) missing mapping from visual elements to underlying concepts, (3) only superficial mapping, and (4) improper mapping. As shown in Appendix Figure 8, these deficiencies collectively lead to poor output. Furthermore, to enable more in-depth analysis through quantitative data, we count proportion of each deficiency type. As shown in Table 3, incorrect recognition accounts for a small proportion, while majority is missing, superficial and improper cross-domain mapping. Therefore, *improving process of linking visual elements to underlying concepts is the key to improving MLLMs performance.*

**Variations across Metaphor Types.** Moreover, we compare MLLMs performance among different video metaphor types. As shown in Figure 5, both close-source and open-sourced MLLMs exhibit significantly lower performance on the latter four types of video metaphor. Generally, videos of the latter four types contain richer metaphorical visual elements, whereas the former four types are relatively simpler. Therefore, *MLLMs perform worse on metaphor types requiring more cross-domain mapping, indirectly supporting that mapping augmentation is the core of improvement.*

## 4. MetaphorBoost

Based on above evaluation and analysis, we find that ineffective cross-domain mapping is the primary factor limiting

current MLLMs performance in metaphorical video understanding. To this end, as illustrated in Figure 6, we first construct a metaphorical knowledge graph as external scaffold, then propose MetaphorBoost, a method that improves MLLMs via inference-time mapping augmentation based on the constructed metaphorical knowledge graph.

### 4.1. Metaphorical Knowledge Graph

Considering metaphor understanding typically needs interconnected linking, we use knowledge graph for augmentation due to its intrinsic multi-hop support. And recognizing the need for metaphorical knowledge beyond general common sense, we construct the first metaphor-specific knowledge graph, containing 54,687 nodes and 200,268 edges.

Specifically, to construct the metaphorical knowledge graph, we first collect public textual metaphorical datasets, which contain extensive real-world metaphorical concept pairs. All texts in datasets are represented as $\mathcal{D} = \{d_1, d_2, \ldots, d_N\}$, where $N$ is amount. Based on this corpus, we use DeepSeek-V3.2 to extract metaphorical concept pairs from each text, which will serve as nodes in knowledge graph, as follows:

$$\mathcal{C} = \bigcup_{i=1}^{N} \text{Extract}(d_i) = \bigcup_{i=1}^{N} \{(c_i^s, c_i^t)\} \qquad (2)$$

where $(c_i^s, c_i^t)$ are the source and target concepts with metaphorical mapping relationship, and $\mathcal{C}$ is the complete set, $|\mathcal{C}| = 54,687$. Then we connect all obtained concepts:

$$\mathcal{G} = (\mathcal{C}, \mathcal{E}), \mathcal{E} = \{(c_i, c_j) \mid c_i, c_j \in \mathcal{C}, \text{ Link}(c_i, c_j) = 1\} \qquad (3)$$

where $\mathcal{G}$ is the metaphorical knowledge graph, $\mathcal{E}$ is the edge set, $|\mathcal{E}| = 200,268$, $\text{Link}(\cdot, \cdot)$ indicates whether existing linking. Detailed textual metaphorical datasets $\mathcal{D}$ are in Appendix D.1. Prompt for extracting is in Appendix D.2.

### 4.2. Inference-time MetaphorVU Boosting

Based on above metaphorical knowledge graph, we develop MetaphorBoost, aiming to consistently improve MLLMs performance via augmenting the cross-domain mapping.

Specifically, to obtain source nodes for performing mapping augmentation, MetaphorBoost first uses given MLLM to comprehensively identify visual elements appearing in the video and output a keyword list $\mathcal{K}$, as illustrated in follows:

$$\mathcal{K} = \text{Identify}(v \oplus t) = \{k_1, k_2, \ldots, k_m\} \qquad (4)$$

where $m$ is the amount of identified keywords in $\mathcal{K}$. Then, MetaphorBoost queries the metaphorical knowledge graph with a maximum of $h$ hops, and retains top-$z$ target nodes that simultaneously link to the most keywords, as following:

$$\mathcal{R} = \text{Top-}z \left( \bigcup_{i=1}^{m} \mathcal{N}_{\mathcal{G}}^{h}(k_i), \text{ deg}(\cdot, \mathcal{K}) \right) \qquad (5)$$

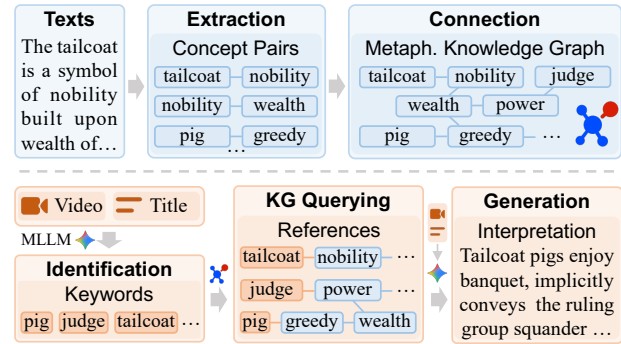

*Figure 6.* We construct a metaphorical knowledge graph and then propose MetaphorBoost, improving MLLMs performance on metaphorical video understanding via mapping augmentation.

where $\mathcal{N}_{\mathcal{G}}^{h}(k_i)$ denotes the nodes within $h$ hops from keyword $k_i$ in metaphorical knowledge graph, $\text{deg}(\cdot, \mathcal{K})$ represents the number of edges linking a target concept to the source keywords, and $\mathcal{R}$ is the resulting set. Finally, with retrieved concepts as reference, MetaphorBoost uses the given MLLM to reveal implicit meanings in thinking $\hat{\tau}$ and finally generate video metaphor interpretation $\hat{o}$, as follows:

$$\hat{\tau}, \hat{o} = \text{Generate}(v \oplus t \oplus \mathcal{R}) \qquad (6)$$

Detailed prompts for process of identifying and generating are shown in Appendix E.1 and Appendix E.2, respectively.

### 4.3. Effectiveness of MetaphorBoost

To extensively validate effectiveness of MetaphorBoost, we conduct experiments on multiple base models, results are in Table 2. For fair comparison, MLLM settings remain consistent with baselines. For method-specific hyperparameters, number $z$ is 10, hops $h$ is 2. Main conclusion is follows. And hyperparameter experiments are in Appendix 6.

**MetaphorBoost can consistently improve MLLMs on metaphorical video understanding.** As shown in Table 2, based on Qwen2.5-VL-7B-Instruct, average score improve from 33.8 to 37.9 by MetaphorBoost, surpassing previous post-training methods. Based on Qwen3-VL-8B-Thinking, average score improve from 52.0 to 55.9, surpassing previous inference-time scaling methods. Based on Gemini-3-Pro, average score improve from 63.8 to 66.1, achieving state-of-the-art score. Overall, mapping augmentation via metaphorical knowledge graph can effectively and consistently boosts MLLMs on metaphorical video understanding.

### 4.4. Ablation of MetaphorBoost

To further explore, we conduct ablation on introducing external knowledge, constructing graph structure, and using metaphor-oriented knowledge in Table 4. Conclusions are:

**External knowledge is important for mapping augmen-**

*Table 4.* Ablation results show that external knowledge is important for mapping augmentation, structured knowledge graph provides more effective augmentation than plain text, and augmentation by metaphor-oriented knowledge outperforms commonsense knowledge.

| Method | Body L. | Atmosph. L. | Cultural S. | Natural. S. | Causal M. | Analog. M. | Surreal N. | Perform. N. | Average |
|---|---|---|---|---|---|---|---|---|---|
| MetaphorBoost (Qwen3-VL-8B-Thinking) (Ours) | **61.8** | **71.0** | **71.8** | **61.3** | **36.7** | **47.1** | **45.7** | 31.5 | **55.9** |
| w/o external augmentation | 57.1 | 69.9 | 67.6 | 60.3 | 33.9 | 44.9 | 40.5 | **36.6** | 53.4 |
| w/o graph-structure augmentation | 60.5 | 70.3 | 69.8 | 61.0 | 30.0 | 43.3 | 45.5 | 30.8 | 54.3 |
| w/o metaphor-oriented augmentation | 57.3 | 67.5 | 65.6 | 61.0 | 30.0 | 46.0 | 42.2 | 30.0 | 52.5 |

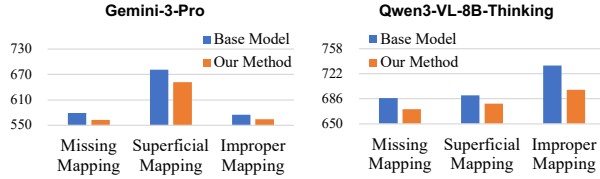

*Figure 7.* Amount of three kinds of bad mapping reduces, proving MetaphorBoost can effectively enhance cross-domain mapping.

**tation.** "w/o external augmentatio" means querying the MLLM itself for augmentation instead of using external knowledge. The performance drops compared to MetaphorBoost, indicating that external knowledge helps compensate for MLLMs deficiency in the cross-domain mapping.

**Knowledge graph provides more effective augmentation than plain text.** "w/o graph-structure augmentation" means retrieving from raw textual metaphorical datasets instead of querying the knowledge graph. The performance drop demonstrates that graph structures provide more effective mapping augmentation by explicit relational connections.

**Metaphor-oriented augmentation outperforms commonsense augmentation.** "w/o metaphor-oriented augmentation" means using ConceptNet[6], a general commonsense knowledge graph, instead of our metaphorical knowledge graph. Performance drops, further supporting MetaphorVU requires the high-order cognition beyond basic knowledge.

### 4.5. Detailed Analysis for MetaphorBoost

**Decline of Bad Mapping Amount.** To further reveal why MetaphorBoost achieves the performance improvement, we analyze thinking process of MetaphorBoost and count the occurrences of missing, superficial and improper mapping, and compare with base models. As shown in Figure 7, the reduced amount of missing, superficial and improper mapping confirms that *MetaphorBoost effectively boosts metaphorical video understanding by enhancing the capability linking visual elements to external underlying concepts.*

**Case Study.** To provide more concrete illustration of reasons why MLLMs struggle with metaphorical video understanding, as well as how MetaphorBoost improves performance, we present a representative case study. As shown in

Appendix Figure 8, the green, orange, and blue highlights indicate missing mapping, superficial mapping, and improper mapping respectively, collectively leading to poor metaphorical video interpretation. And *MetaphorBoost effectively mitigates the three types of deficiencies, thereby improving MLLMs performance on metaphorical video understanding.*

## 5. Related Work

**Metaphor Understanding.** Prior research on metaphor understanding primarily focuses on text and images, with video metaphor remaining relatively scarce. For textual metaphor, works aim to detect metaphor based on relationships between tokens, and to identify the source and target domains (Prystawski et al., 2023; Tian et al., 2024; Zheng et al., 2025b). For image metaphor, some works collect images such as internet memes for datasets (Xu et al., 2022; Yang et al., 2025b; Kundu et al., 2025; Saakyan et al., 2025; Chakrabarty et al., 2022), or explore multimodal fusion to improve performance (Qian et al., 2025; Zheng et al., 2025a; Xu et al., 2024). Compared to text and images, videos are temporal and convey richer information, more likely containing complex metaphor. Recently, a few studies advance video metaphor research by constructing datasets from advertisement videos (Kalarani et al., 2024; Jia et al., 2025; Long et al., 2025; Zhang et al., 2025b). However, these are limited to the advertising domain, which may not accurately reflect the capabilities in complex real-life scenarios.

**Deep-semantic Video Understanding.** With the advancement of MLLMs, recent work begins to explore deep-level video understanding beyond basic object recognition or event description. Some studies present scientific experiment in videos and require to predict outcomes (Deng et al., 2025), illustrate complex domain knowledge and require to solve new problems not shown in the video (Hu et al., 2025), show incomplete event and ask to infer the underlying logic of event (Chen et al., 2025), and display objects from the same scene across separate frames, requiring to reason about spatial relationships and motion trajectories (Swetha et al., 2025; Yang et al., 2025a). Additionally, some studies investigate advertisement video understanding, as discussed in above paragraph. Overall, research on deep-semantic video understanding remains in the early stages. Our work contributes to this direction by systematically introducing metaphorical video understanding as a new challenging task.

---

[6]https://huggingface.co/spaces/cstr/conceptnet_db

Recently, **MMR-V** is proposed to evaluate the implicit reasoning in video understanding (Zhu et al., 2026), which is a highly valuable related work. Upon careful comparison, the core unique value of our work lies in the systematicness and depth on metaphorical video understanding compared with MMR-V. Specifically, MMR-V aims to assess a broad spectrum of reasoning abilities, where metaphor-related content appears as one of many test scenarios rather than a dedicated focus. In contrast, our work focuses specifically on metaphorical video understanding. We construct a systematic taxonomy of video metaphor and carefully curate a benchmark spanning diverse metaphor types and topics, thereby enabling more comprehensive and fine-grained analysis of MLLMs' metaphorical video understanding capability. From a broader perspective, our work and MMR-V can complement each other, jointly enabling a deep evaluation of high-order cognitive capabilities to improve MLLMs.

**Multimodal Sarcasm.** Multimodal sarcasm research is relevant to our work and deserves discussion. In general, multimodal sarcasm understanding and metaphorical video understanding differ in their core capability requirements and the types of implicit meanings they encompass. In terms of core capability requirements, sarcasm primarily relies on identifying apparent contradictions among elements (Zhuang et al., 2025; Wang et al., 2025c), whereas metaphorical video understanding requires models to perform cross-domain mapping, i.e., linking visual elements to underlying concepts. In terms of implicit meanings, sarcasm mainly focuses on conveying critical and negative thoughts (Wang et al., 2025a; Ou & Li, 2025), whereas metaphorical video understanding covers a broader and more diverse range of implicit meanings, as in Figure 2, encompassing various forms prevalent in everyday life.

## 6. Conclusion

In this paper, to fill the gap in prior research on metaphorical video understanding, we design the first systematic video metaphor taxonomy and construct MetaphorVU-Bench, enabling a comprehensive evaluation of metaphorical video understanding. Extensive experiments reveal that current MLLMs struggle with accurate metaphorical video understanding, primarily due to defective cross-domain mapping. Motivated by these findings, we construct a metaphorical knowledge graph and propose MetaphorBoost, which can consistently improve MLLM performance via mapping augmentation. This paper offers a promising direction for MLLM advancement and can inspire further research.

## Impact Statement

This paper presents work whose goal is to advance the field of machine learning. There are many potential societal consequences of our work, none of which we feel must be specifically highlighted here.

## Acknowledgments

We sincerely thank the reviewers for their insightful comments and valuable suggestions. This work was supported by the National Key R&D Program of China (2024YFC3308000), the Natural Science Foundation of China (No. 62536008, 62476265, 62306303).

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

## A. Theoretical Basis for Video Metaphor Taxonomy

To ensure reliable and principled evaluation, a systematic video metaphor taxonomy is essential for building the benchmark. Since no prior works have explored this kind of taxonomy, we draw on multimodal metaphor theory (Forceville et al., 2009; Forceville & Urios-Aparisi, 2009) and its extensions in the video field (Bordwell, 2013b; Stam, 2017; Schechner, 2017; Chandler, 2022), designing the first systematic video metaphor taxonomy, the details are illustrated in follows:

According to Film Mise-en-scène Theory (Bordwell et al., 2004; Gibbs & Gibbs, 2002; Arnheim, 1957), video metaphors can be realized through visual element arrangement within frames. **Body Language** corresponds to Performance Staging—physical movements, facial expressions, and postures serve as metaphorical source domains, mapping abstract emotional states onto visible bodily behaviors (Naremore, 1988; Gibbs & Gibbs, 2002). **Atmosphere Language** corresponds to Environmental Staging—color tones, lighting, and composition serve as metaphorical carriers of emotional tone (Arnheim, 1957; Bellantoni, 2012; Brown, 2016).

According to Symbol and Symbolism Theory (Rawls, 1999; Jung, 2012; Eliade, 1991; Chandler, 2022), video metaphors can be realized through symbolic signs carrying conventional or archetypal meaning. **Cultural Symbol** corresponds to conventionally established symbols within specific cultural contexts—their meaning depends on cultural knowledge (Danesi, 2018; Yeats, 1998). **Naturalistic Symbol** corresponds to natural elements with universal symbolic meaning rooted in shared human experiences and collective unconscious (Jung, 2012; Campbell, 2008; Ferber et al., 1999).

According to Montage Theory (Eisenstein, 2018; Kuleshov & Kuleshov, 1974; Pudovkin, 2013; Cutting, 2016), video metaphors can be realized through dialectical collision between shots. **Causal Montage** corresponds to causal reasoning—temporal shot juxtaposition implies causal relationships, with audiences automatically completing causal chains (Pudovkin, 2013; Bordwell, 2013a; Carroll, 1996). **Analogical Montage** corresponds to analogical reasoning—juxtaposition of unrelated shots guides audiences to identify structural similarities and establish cross-domain mappings (Eisenstein, 2018; Whittock, 1990; Fauconnier & Turner, 2008).

According to Theatre Semiotics and Performance Theory (Elam, 2003; Schechner, 2017), narrative-based video metaphors operate through distinct semiotic registers. **Surreal Narrative** employs what terms "virtual performance"—animated or AI-generated characters transcend physical constraints, enabling metaphorical expression through impossible actions, fantastical transformations, and dreamlike scenarios that would be unachievable in reality (Auslander, 2022; Manovich, 2002; Wells, 2013). **Performative Narrative** relies on embodied performance where human actors serve as direct meaning carriers; audiences decode metaphorical connotations through theatrical conventions such as exaggerated expressions, symbolic staging, and dramatized conflicts (Schechner, 2017; Elam, 2003).

## B. Multi-stage Filtration Prompts and Manual Annotation Guideline

### B.1. Prompt for LLM Filtration

To efficiently isolate metaphorical videos from billions of videos, we first use a powerful LLM (GPT-5) to analyze the video introduction, automatic speech recognition (ASR) result and audience comments to determine whether each video contains metaphorical logic, the detailed prompt is shown in Figure 9.

### B.2. Prompt for MLLM Filtration

Considering above filtration process does not directly use visual information and LLM analysis may not align with the actual video, to conduct further check and filtration, a powerful MLLM (Gemini-3-Pro) is used to verify whether above analysis is consistent with original videos, the detailed prompt is shown in Figure 10.

### B.3. Prompt for Human Filtration

Then, a human team performs final filtration based on the original video, video introduction and audience comments, resulting in 860 videos with definite metaphorical logic. Additionally, annotators identify the metaphor type for each video, balancing the number of samples across each metaphor type as much as possible. The detailed prompt is shown in Figure 11.

*Table 5.* Details of metaphorical textual datasets.

| Name | URL | # Samples |
|------|-----|-----------|
| Manual_Metaphors | `https://huggingface.co/datasets/Sasidhar1826/manual_data_on_metaphors` | 718 |
| Metaphor_Novelty | `https://huggingface.co/datasets/omarmomen/metaphor-novelty` | 200 |
| Metaphor_Explanation | `https://huggingface.co/datasets/JasonShao/Chinese_Metaphor_Explanation` | 28000 |
| Metaphor_Dataset | `https://huggingface.co/datasets/liyucheng/chinese_metaphor_dataset` | 8030 |

## B.4. Manual Annotation Guideline

When annotating video metaphor interpretation, we require human annotators to reference video introduction and audience comments and follow a fixed format (i.e., *specifying which visual elements convey which implicit meanings*). The detailed guideline is shown in Figure 12.

## C. Prompt for Evaluation and LLM Judge, and Consistency Experiments

### C.1. Prompt for Evaluation

Generally, MLLMs first recognize visual contents, establish projection to external concepts and unveil implicit meanings in thinking process, then interpret which visual contents convey which implicit meanings in final output. Details of evaluation prompt are in Figure 13.

### C.2. Prompt for LLM Judge

Since the output video metaphor interpretation in MetaphorVU-Bench is free-form text, rule-based metrics are difficult to provide a score aligning with actual human habits (Mayfield et al., 2024; Li et al., 2025d). To this end, we follow the metrics in previous free-form QA evaluation works (Li et al., 2025b;e; Yu et al., 2025; Long et al., 2025), using DeepSeek-V3.2 as LLM judge. Detailed prompt for LLM judge are in Figure 14.

### C.3. Consistency Experiments for LLM Judge

To verify the reliability of the LLM judge, we randomly sample 100 instances from the evaluation results and have human annotators score the model-generated video metaphor interpretations following the same evaluation guidelines. We then analyze the consistency between human scores and LLM judge scores. The results show a Pearson correlation coefficient of 0.85 with a p-value of 3e-20 ($p < 0.001$), indicating a strong positive correlation with high statistical significance between human and LLM judgments. This validates the reliability and effectiveness of using LLM as an automatic judge in our framework.

## D. Textual Datasets and Prompt in Metaphorical KG Construction

### D.1. Details of Metaphorical Textual Datasets

To construct a metaphorical knowledge graph, we first collect textual metaphorical datasets, which contain extensive metaphorical concept pairs. The details of used textual metaphorical datasets are shown in Table 5. Note that a portion of the data was originally in Chinese, to ensure the universality of the metaphorical knowledge graph, we use GPT-5 to translate the original text into English.

### D.2. Prompt for Extracting Metaphorical Concept Pairs

Since several previous works that have been widely recognized by the community have demonstrated that current LLMs possess excellent information extraction capabilities (Tang et al., 2023; Li et al., 2024; Tang et al., 2024; Li et al., 2025a), we adopt the same approach and use DeepSeek-V3.2 to extract metaphorical concept pairs from each text, which will serve as nodes in the knowledge graph. The specific prompt is shown in Figure 15.

# E. Prompts for Identification and Generation in MetaphorBoost

### E.1. Prompt for Identifying Visual Elements

At the time of MLLMs inference, to obtain the source nodes for performing cross-domain mapping augmentation, Metaphor-Boost first uses the given MLLM to comprehensively identify visual elements appearing in the video and output a keyword list. The specific prompt is shown in Figure 16.

### E.2. Prompt for Generating Video Metaphor Interpretation

Based on above identifying results, MetaphorBoost queries the metaphorical knowledge graph. And then with retrieved concepts as augmentation, MetaphorBoost uses the given MLLM to unveil implicit meanings and finally generate video metaphor interpretation. The specific prompt is shown in Figure 17.

# F. Details of Reasoning-based Baselines

Reasoning-enhanced Methods improve the reasoning ability of base model by post-training or inference-time scaling, this type of baseline includes 7 methods:

**VideoRFT** (Wang et al., 2025b) is a reinforcement fine-tuning approach designed to cultivate video reasoning capabilities in multimodal large language models. It follows a two-stage training scheme: supervised fine-tuning with chain-of-thought annotations, followed by reinforcement learning with a semantic-consistency reward to promote alignment between textual reasoning and visual evidence. While VideoRFT achieves strong performance on various video reasoning benchmarks, it primarily focuses on foundational cognitive tasks such as object recognition and event understanding, limiting its capability for metaphorical video understanding.

**Vision-R1** (Huang et al., 2025) aims to enhance multimodal reasoning capability through reinforcement learning inspired by DeepSeek-R1. It constructs a 200K multimodal CoT dataset via modality bridging and data filtering, and employs Progressive Thinking Suppression Training to refine complex reasoning ability. However, similar to VideoRFT, it is primarily tailored for low-level video understanding tasks involving logical and mathematical reasoning, rather than the cross-domain mapping required for metaphorical video interpretation.

**ReAd-R** (Long et al., 2025) is a reinforcement learning model specifically designed for advertisement video understanding, targeting tasks that require perceiving beyond objective physical content, such as marketing logic and persuasive strategies. Compared to VideoRFT and Vision-R1, ReAd-R is more relevant to our task as advertisement videos often contain implicit meanings. However, its domain-specific training limits generalizability to broader metaphorical video understanding.

**LTR** (Liao et al., 2025) (Language-centric Tree Reasoning) enhances video question-answering through structured logical reasoning at inference time. It recursively divides complex cognitive questions into manageable parts and performs bottom-up reasoning within a language-centric logical tree. While LTR improves reasoning transparency on various video QA benchmarks, its structured decomposition approach may not effectively capture the cross-domain mapping required for understanding video metaphors.

**ViTCoT** (Zhang et al., 2025a) (Video-Text Interleaved Chain-of-Thought) introduces a video reasoning paradigm that interleaves visual and textual information during reasoning, enabling models to re-examine visual content while reasoning. Although ViTCoT improves general video understanding by better integrating visual modality, it still focuses on explicit content reasoning rather than cross-domain mapping required in metaphorical understanding.

**Prompt Engineering** (Wei et al., 2022) refers to chain-of-thought prompting, which improves reasoning ability by generating intermediate reasoning steps through carefully designed prompts. In our experiments, we design prompts that explicitly encourage the model to perform cross-domain mapping from visual contents to implicit meanings, representing a straightforward baseline for metaphorical video understanding.

**Few-shot Example** (Dong et al., 2024) is based on in-context learning, where models make predictions based on contexts augmented with demonstration examples. For metaphorical video understanding, we provide annotated examples demonstrating how to project explicit visual contents onto abstract concepts. Together with Prompt Engineering, this represents the most direct approach for adapting existing models to our task.

*Table 6.* Experiments about query strategy and hyperparameters.

| Method | Body L. | Atmosph. L. | Cultural S. | Natural. S. | Causal M. | Analog. M. | Surreal N. | Perform. N. | Average |
|---|---|---|---|---|---|---|---|---|---|
| MwtaphorBoost (Qwen3-VL-8B-Thinking) | **61.8** | 71.0 | 71.8 | 61.3 | **36.7** | 47.1 | **45.7** | 31.5 | **55.9** |
| w/o common connection | 59.5 | 69.7 | **72.3** | 62.2 | 35.0 | 45.3 | 43.7 | 33.5 | 54.8 |
| w/ hop $h = 1$, return $z = 10$ | 59.3 | **73.0** | 68.5 | **65.4** | 25.3 | 46.4 | 42.9 | 32.5 | 54.5 |
| w/ hop $h = 2$, return $z = 5$ | 60.1 | 71.8 | 70.0 | 63.7 | 31.3 | **47.5** | 45.1 | **35.6** | 55.7 |

## G. Experiments about Query Strategy and Hyperparameters

In the inference-time mapping augmentation, MetaphorBoost queries the metaphorical knowledge graph with a maximum of $h$=2 hops, and retains the Top-$z$=10 target nodes that are simultaneously associated to the most keywords, thereby maximizing the advantages of the knowledge graph, namely its support for multi-hop and structured reasoning. To convincingly demonstrate the effectiveness of this query strategy, we conduct further experiments, as shown in Table 6.

The setting "w/o common connection" means that instead of retaining results that simultaneously have as many connections to the query keywords as possible, results are retained randomly. The experimental results show that the average performance decreases. This, to some extent, *demonstrates the advantages of using a knowledge graph, which can provide low-noise augmentation via structured federated query*.

Furthermore, to provide a deeper investigation into the underlying mechanism of MetaphorBoost, we conduct experiments on its two key hyperparameters: the maximum number of hops $h$ for querying the knowledge graph and the number of retained results $z$, with default values of 2 and 10, respectively. In the table, we present results for $h = 1$ and $z = 5$. The experimental results show that while performance fluctuates across different subsets, the average scores of all variants remain lower than those of MetaphorBoost with default settings. This further validates the effectiveness of leveraging the knowledge graph for cross-domain mapping—demonstrating that *the knowledge graph can provide effective, reasonably deep, and low-noise augmentation for metaphor interpretation*.

## H. More Examples of MetaphorVU-Bench

We provide more examples for all eight video metaphor types, specifically, Body Language is in Figure 18, Atmosphere Language is in Figure 19, Cultural Symbol is in Figure 20, Naturalistic Symbol is in Figure 21, Causal Montage is in Figure 22, Analogical Montage is in Figure 23, Surreal Narrative is in Figure 24, Performative Narrative is in Figure 25.

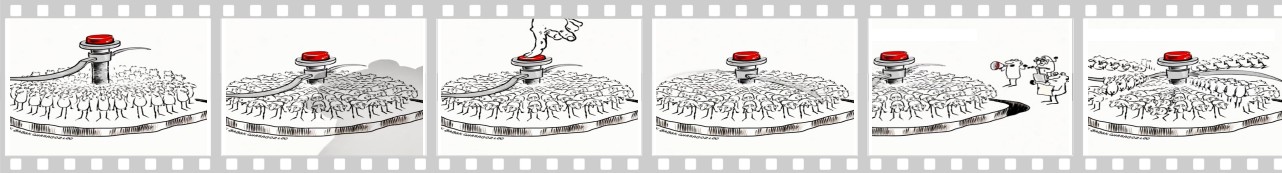

**Base Model**

**MetaphorBoost**

# Thinking Process
The video contains a rotating scythe machine, people bowing their heads, then a leader encouraging resistance but getting sacrificed, and finally everyone standing together to jam the machine. In this video, the scythe machine represents danger, and then the giant hand represents a protective force, and the collective standing represents teamwork. Therefore, the video expresses that when facing danger, there is a force to protect everyone, and teamwork is also important.

# Video Metaphor Interpretation
· Rotating scythe machine in the middle of the crowd, conveys a sense of danger and threat to the people around it
· Giant hand pressing down on the machine, conveys a powerful protective force watching over the crowd
· People collectively standing up and jamming the machine with their bodies, conveys that teamwork and cooperation can help overcome difficult challenges.

# Thinking Process
The video contains a rotating scythe machine, some people bowing their heads, a leader encouraging resistance but sacrificed, finally everyone standing together to jam the machine. The scythe machine symbolize authoritarian oppression and in video the people bowing symbolizes forced submission and self-degradation, and then giant hand symbolize the evil exploiting class, the collective standing symbolizes united resistance against oppression. Therefore, the video expresses that under authoritarian rule, people are forced to submit; individual resistance, though brave, inevitably leads to sacrifice; only collective unity can truly dismantle the oppressive regime.

# Video Metaphor Interpretation
· Rotating scythe machine and giant hand pressing down, conveys authoritarian rule forcing submission through violence.
· Crowd bowing to avoid the scythe, conveys people having to degrade themselves to survive.
· Everyone standing together to jam the machine, conveys that only collective unity can dismantle the oppressive regime.

*Figure 8.* The green, orange, and blue highlights indicate missing mapping, superficial mapping, and improper mapping respectively, these deficiencies collectively lead to poor metaphorical video interpretation. MetaphorBoost effectively mitigates the three types of deficiencies, thereby improving MLLMs performance on metaphorical video understanding.

---

**Prompt for LLM Filtration**

prompt = \
"""You are an analyst proficient in visual communication, linguistics, and social media content, skilled at interpreting the deeper meanings of videos through user comments.

#### Task
Based on the provided video comment section, determine whether the video likely uses metaphor to express ideas or emotions. You will analyze solely based on the comment section content, without considering the video title or any text within the video itself.

#### Definition of "Video Metaphor"
In this task, a "video metaphor" refers to video content (as inferred through textual cues) that does not directly or literally express its core idea, but instead expects viewers to understand the implied meaning through the visual content presented. Since you are only analyzing the comment section, you need to infer from the comments whether viewers have perceived a metaphor.

#### Analysis Steps and Key Questions
Please analyze based on the comment section content, referring to the following key questions:

- Are commenters asking about the meaning of the video? (e.g., "What does this mean?" "I don't understand")
- Are commenters actively sharing their own interpretations? (e.g., "This is about the rat race, right?" "I see my former self in this")
- Do the comments contain numerous abstract terms not directly related to the video's surface content? (For example, if the video shows a stone, but the comments discuss "perseverance," "life," or "staying true to oneself")
- Is there a consensus on a deeper interpretation? That is, multiple comments pointing to the same metaphorical understanding.
- Do the comments indicate that viewers are contemplating the symbolic meaning of the video rather than its literal content?

#### Output Format
You must strictly follow the JSON format below and add no other content:

```json
{{
  "reasoning": "Briefly explain your analysis process based on the comment section. For example: 'Multiple commenters mentioned associations with 'rat race' and 'workplace pressure,' and shared personal experiences, suggesting the video likely uses metaphor.' Or: 'The comments mainly focus on praising performance techniques or surface-level content, with no deeper interpretations observed, so there is likely no metaphor.'",
  "is_metaphor": 1 or 0
}}
```

---

### Now, please analyze the following video information:

Video introduction: {introduction}
Video ASR result: {asr_result}
Video comment section: {comments}

"""

*Figure 9.* Prompt for LLM filtration.

---

**Prompt for MLLM Filtration**

{video}

prompt = \
"""You are an expert proficient in visual communication, multimedia content analysis, and metaphor interpretation, skilled at validating the reasonableness of linguistic analysis through a video's visual elements.

#### Task
Given a video and the first-stage metaphor analysis reasoning based on audience comments, please determine whether this reasoning is reasonable. That is, based on the video content itself (visual elements such as scenes, objects, actions, etc.), assess whether the first-stage reasoning (interpretation based on comments) is credible and supported. Your analysis should focus on the consistency between the video's visual content and the first-stage reasoning, without relying on comments or external information.

#### Definition of "Reasonableness"
In this task, "reasonableness" refers to whether the metaphor inferred from comments in the first-stage reasoning is consistent with the video's visual content. If the video's visual elements clearly support or imply the metaphorical interpretation in the reasoning, it is considered reasonable; if the video content expresses meaning directly and literally, contradicts the metaphor, or the visual elements cannot support the abstract interpretation, it is considered unreasonable.

#### Analysis Steps and Key Questions
Please analyze based on the video content, referring to the following key questions:
- Do the video's visual elements (such as scenes, objects, character actions, colors, composition) directly or indirectly support the metaphorical interpretation mentioned in the first-stage reasoning?
- Does the video have a clear, literal meaning that conflicts with the metaphorical interpretation?
- Do the abstract concepts inferred from audience comments (such as "life," "perseverance," "rat race") have corresponding symbols or implications in the video's visuals?
- Does the video contain ambiguous or polysemous elements that make the metaphorical interpretation plausible, or is it obviously just surface-level content?
- Is the first-stage reasoning possibly based on biases from comments or external knowledge rather than the video itself?
- If the video content is actually just text or chat logs and similar formats, these do not qualify as videos and should be directly excluded.

#### Output Format
You must strictly follow the JSON format below and add no other content:

```json
{{
  "multimodal_check": "Briefly explain your analysis process based on the video's visual content. For example: 'The video visually shows a stone rolling through wind and rain, symbolizing perseverance and resilience, which is consistent with the 'life metaphor' in the first-stage reasoning, therefore reasonable.' Or: 'The video content is a straightforward tutorial demonstration with no abstract elements, having no visual connection to the 'rat race metaphor' in the first-stage reasoning, therefore unreasonable.'",
  "is_reasoning_valid": 1 or 0
}}
```

---

### Now, please analyze the following video information:

First-stage reasoning: {reasoning}
"""

*Figure 10.* Prompt for MLLM filtration.

---

**Prompt for Human Filtration**

# Background
Constructing a metaphor video understanding benchmark to evaluate VLMs' metaphor comprehension capabilities.

**Evaluation Method:** Input a short video, and expect the VLM to interpret the implicit ideas expressed through the video's actual presented content.

---

# Requirements

## Data Format

The provided data includes the following content:

- **Video ID** - Open the platform and enter the video ID to view the corresponding video (e.g., 74416621513)

- **LLM Initial Analysis** - The initial analysis uses user comments as the basis to analyze the video's metaphorical logic (e.g., "The comment section contains numerous comments expressing deep emotional resonance with themes of love, regret, fate, and life philosophy, such as 'love that cannot be obtained, never to meet again,' 'do you have regrets,' 'love constrained by identity,' 'like fireworks, blooming then fading,' etc. Commenters actively share personal emotional experiences and abstract interpretations, indicating that viewers generally perceive the video content as having symbolic meaning rather than staying at the surface narrative level. Multiple comments point to a consensus metaphorical understanding of 'regret in love' and 'impermanence of fate.'")

- **MLLM Secondary Analysis** - The secondary analysis uses the complete video and initial analysis as input, employing VLM to conduct multimodal verification and further analysis of the video's metaphorical logic (e.g., "The video contains multiple visual elements such as figures running in the rain, blooming fireworks, withered roses, and solitary sitting postures. These elements are commonly used to symbolize abstract concepts such as the brevity of love, regret, and impermanence of fate. For example, fireworks symbolize brief yet brilliant love, withered roses symbolize the fading of love, and running in the rain symbolizes emotional struggle or escape. These visual symbols are highly consistent with the metaphors mentioned in the comments such as 'love that cannot be obtained,' 'regret,' and 'impermanence of fate.' The video content itself supports these abstract interpretations.")

---

## Annotation Requirements

### (a) Yes/No Question
**Does the video contain metaphor?** That is, does it implicitly express certain ideas or emotions through the actually presented content?
- If **"Yes"**: Continue to annotate (b) and (c)
- If **"No"**: No further annotation needed

### (b) Multiple-Choice Question (Select All That Apply)
**Metaphor Type:** What is the primary method through which the video achieves metaphor?

**8 Candidate Options:**

| Type | Description |
|------|-------------|
| **Body Language** | Video conveys implicit meanings through character body movements, typically some exaggerated or semantically meaningful actions. |
| **Atmosphere Language** | Video conveys implicit meanings through environmental atmosphere, such as variations in color, lighting, and composition. |
| **Cultural Symbol** | Video conveys implicit meanings through symbolism of cultural artifacts, such as flying Chinese Kongming lanterns or building a Christianity cross. |
| **Naturalistic Symbol** | Video conveys implicit meanings through symbolism of natural elements, such as animal behaviors, plant growth, and changing starry skies. |
| **Causal Implication** | Video conveys implicit meanings through causal montage editing, which can guide audiences to infer some causal logic in their brain. |
| **Analogical Implication** | Video conveys implicit meanings through analogical montage editing, which can guide audiences to infer analogical logic in their brain. |
| **Synthetic Drama** | Video conveys implicit meanings through drama performed by virtual characters, such as animated cartoons and AI-generated videos. |
| **Human-action Drama** | Video conveys implicit meanings through drama performed by human actors, such as short plays on many short video platforms. |

*Figure 11.* Prompt for Human filtration.

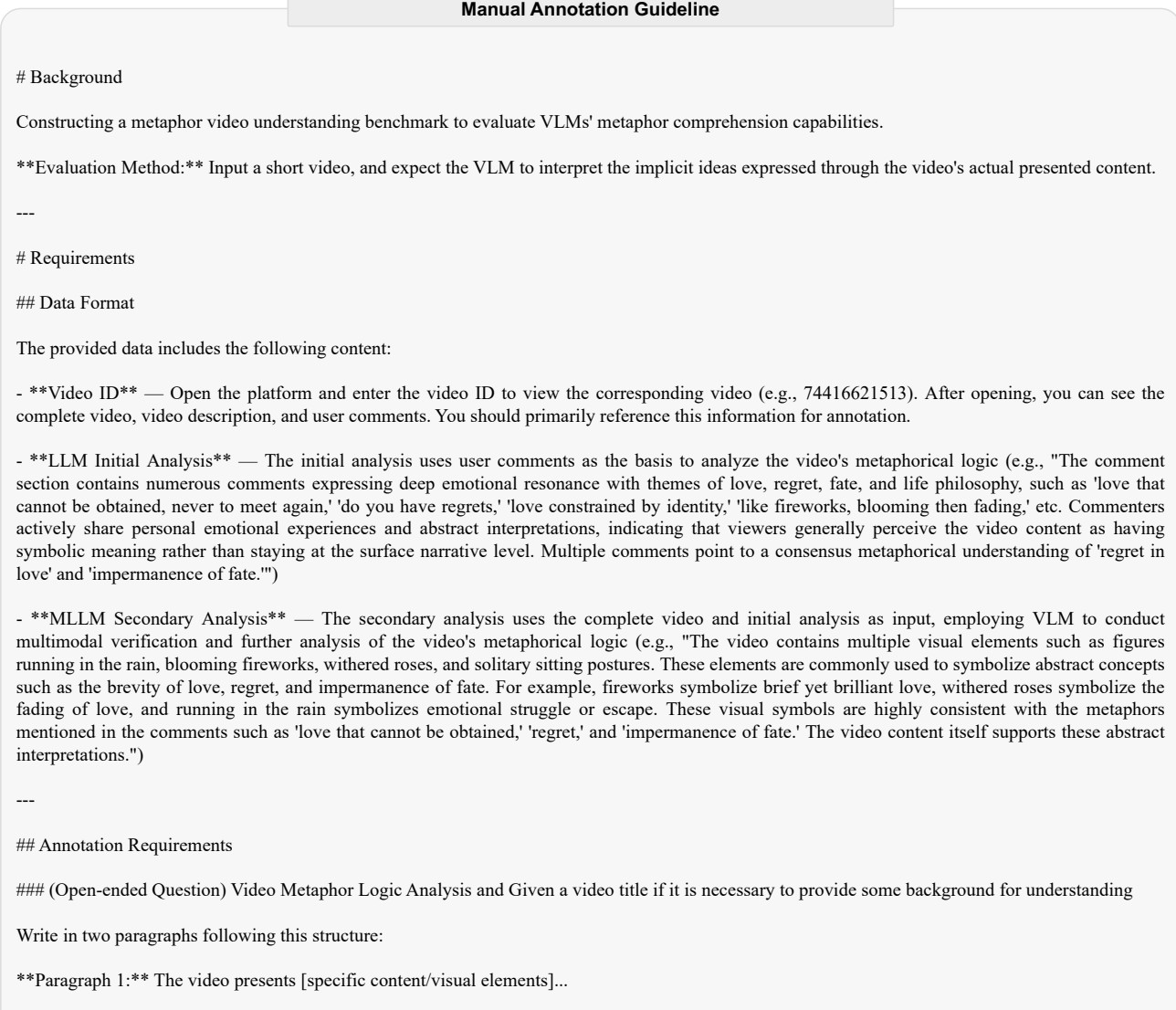

# Background

Constructing a metaphor video understanding benchmark to evaluate VLMs' metaphor comprehension capabilities.

**Evaluation Method:** Input a short video, and expect the VLM to interpret the implicit ideas expressed through the video's actual presented content.

---

# Requirements

## Data Format

The provided data includes the following content:

- **Video ID** — Open the platform and enter the video ID to view the corresponding video (e.g., 74416621513). After opening, you can see the complete video, video description, and user comments. You should primarily reference this information for annotation.

- **LLM Initial Analysis** — The initial analysis uses user comments as the basis to analyze the video's metaphorical logic (e.g., "The comment section contains numerous comments expressing deep emotional resonance with themes of love, regret, fate, and life philosophy, such as 'love that cannot be obtained, never to meet again,' 'do you have regrets,' 'love constrained by identity,' 'like fireworks, blooming then fading,' etc. Commenters actively share personal emotional experiences and abstract interpretations, indicating that viewers generally perceive the video content as having symbolic meaning rather than staying at the surface narrative level. Multiple comments point to a consensus metaphorical understanding of 'regret in love' and 'impermanence of fate.'")

- **MLLM Secondary Analysis** — The secondary analysis uses the complete video and initial analysis as input, employing VLM to conduct multimodal verification and further analysis of the video's metaphorical logic (e.g., "The video contains multiple visual elements such as figures running in the rain, blooming fireworks, withered roses, and solitary sitting postures. These elements are commonly used to symbolize abstract concepts such as the brevity of love, regret, and impermanence of fate. For example, fireworks symbolize brief yet brilliant love, withered roses symbolize the fading of love, and running in the rain symbolizes emotional struggle or escape. These visual symbols are highly consistent with the metaphors mentioned in the comments such as 'love that cannot be obtained,' 'regret,' and 'impermanence of fate.' The video content itself supports these abstract interpretations.")

---

## Annotation Requirements

### (Open-ended Question) Video Metaphor Logic Analysis and Given a video title if it is necessary to provide some background for understanding

Write in two paragraphs following this structure:

**Paragraph 1:** The video presents [specific content/visual elements]...

**Paragraph 2:** ...which implicitly expresses [specific ideas or emotions].

*Figure 12.* Manual annotation guideline.

**Prompt for Evaluation**

{video}
{video title}

prompt = \
"""<< Instruction >>
Analyze the metaphorical logic in this video, i.e., what ideas are implicitly expressed through the content presented.

<< Requirements >>
(1) Thoroughly identify all video content that contains metaphors.
(2) Analyze the underlying ideas of the metaphors deeply and accurately.
(3) Avoid baseless assumptions or forced interpretations.
(4) If there are multiple elements in the video that contain metaphorical logic, list them separately, with each entry as a concise sentence.
(5) Starting with sufficient reasoning, and final output in JSON format as a dictionary, begin with '```json' and end with '```'. The only key is "analysis_dict", in which each analysis entry should follow the sentence structure: "The video presents *** content, implicitly expressing *** idea."

<< Output Format >>
```json
{
"analysis_dict":
{
"analysis_1": "The video presents the *** content, implicitly expressing the *** idea",
"analysis_2": "The video presents the *** content, implicitly expressing the *** idea"
...
}
}
```"""

*Figure 13.* Prompt for evaluation.

**Prompt for LLM Judge**

```
prompt = \
"""<<Task>>
You are an evaluation expert responsible for scoring metaphor interpretation generated by a VLM (Vision-Language Model). The metaphor interpretation typically explains "which video contents implicitly convey which underlying meanings".

The evaluation involves two scoring criteria with different strictness levels:
1. **Strict Score**: A binary score (0 or 1). Requires the interpretation to be both COMPLETE and ERROR-FREE — must cover all major metaphorical meanings from the golden analysis AND contain no significant errors or contradictions. Only award 1 if both conditions are fully satisfied.
2. **Loose Score**: A graded score (0 to 10). Requires only CORE CORRECTNESS — as long as the main interpretation direction is correct, minor omissions or small errors are acceptable.

<<Model-generated Interpretation>>
{model_analysis}

<<Golden Interpretation>>
{golden_analysis}

<<Scoring Guidelines>>

---

**I. Strict Score (Binary: 0 or 1)**

A strict binary assessment of whether the interpretation meets high standards.

| Score | Criteria |
|-------|----------|
| 0 | The interpretation fails to meet EITHER condition: (1) missing any major metaphorical meaning from the golden analysis, OR (2) containing any significant error or contradiction |
| 1 | The interpretation meets BOTH conditions: (1) covers ALL major metaphorical meanings from the golden analysis, AND (2) contains NO significant errors or contradictions |

**Key Principle**: This is a strict pass/fail evaluation. Any notable omission of major content OR any significant error should result in a score of 0. Only a comprehensive and accurate interpretation earns a score of 1.

---

**II. Loose Score (Graded: 0 to 10)**

A lenient assessment focusing on whether the core metaphorical meaning is captured correctly.

| Score | Criteria |
|-------|----------|
| 0 | Completely misses the core meaning, interpretation direction is fundamentally wrong |
| 1-2 | Barely touches the core meaning, interpretation direction is largely incorrect or confused |
| 3-4 | Shows some understanding of the metaphor but the core meaning is only partially correct or somewhat off-track |
| 5-6 | Captures the general direction of the core meaning, but with noticeable gaps in understanding; interpretation is on the right track but imprecise |
| 7-8 | Correctly captures the core metaphorical meaning, interpretation direction is accurate; minor omissions or small errors do not affect this score |
| 9-10 | Clearly and accurately captures the core meaning with good precision; interpretation demonstrates solid understanding of the main metaphor |

**Key Principle**: Focus on whether the main interpretation direction is correct. Minor omissions, small errors, or incomplete coverage should NOT significantly affect the score as long as the core meaning is captured.

---

<<Scoring Procedure>>

**Step 1: Identify Core vs. Supporting Elements**
- Identify the CORE metaphorical meaning from the golden analysis (the central message/theme)
- Identify SUPPORTING elements (specific visual details, secondary meanings, elaborations)

**Step 2: Evaluate for Strict Score**
- Check: Are ALL major metaphorical meanings covered?
- Check: Are there ANY significant errors or contradictions?
- If BOTH conditions are satisfied → Strict Score = 1
- If EITHER condition fails → Strict Score = 0

**Step 3: Evaluate for Loose Score**
- Focus primarily on core meaning correctness
- Be tolerant of omissions and minor errors
- Assign a graded score (0-10) based on how well the core meaning is captured

---

<<Important Notes>>

1. **Semantic Equivalence**: Focus on semantic essence during evaluation; exact wording match is not required. Content with different expressions but the same meaning should be considered a match.

2. **Score Independence**: The two scores evaluate different aspects. A model might score 0 on Strict (due to one missing element or one error) but still score high on Loose (if core meaning is correct).

3. **Reasonable Extensions**: If the model output contains content not mentioned in the golden analysis but is genuinely reasonable and grounded, do not consider it as an error.

4. **Definition of "Major" vs "Minor"**:
- Major elements: Central themes, primary metaphorical mappings, key messages
- Minor elements: Specific details, secondary interpretations, elaborations
- For Strict Score: Missing major elements → 0
- For Loose Score: Missing minor elements → minimal impact

5. **Definition of "Significant Error"**:
- Significant: Contradicts the golden analysis, misinterprets the core meaning, or introduces clearly wrong information
- Minor: Slightly imprecise wording, over-elaboration that doesn't contradict the main meaning

---

<<Output Format>>

Please output strictly in the following JSON format, starting with ```json and ending with ```:

```json
{{
"reasoning": "Your reasoning process, including: 1) Identification of core meaning and major elements from golden analysis; 2) Completeness check for Strict Score; 3) Error check for Strict Score; 4) Core meaning correctness evaluation for Loose Score; 5) Justification for both scores",
"strict_score": score (0 or 1),
"loose_score": score (integer from 0-10)
}}
```"""
```

*Figure 14.* Prompt for LLM judge.

---

**Prompt for Extracting Metaphorical Concept Pairs**

prompt = """You are a semantic association relation extraction expert. Your task is to extract "association pairs" from text—concept pairs that have deep semantic connections.

## Core Concepts
- **Explicit Concept**: Specific words or phrases that explicitly appear in the text (usually concrete and perceivable)
- **Implicit Concept**: The abstract concept that the explicit concept truly points to or implies in the current context

## Extraction Rules
1. Explicit concepts must be words or phrases that explicitly appear in the text
2. Implicit concepts are what the explicit concept deeply points to in the current context (may appear in the text or may need to be inferred)
3. A text may contain zero to multiple association pairs
4. **Output Granularity Requirement**: Both explicit and implicit concepts should be **single words or phrases of at most two words**
   - ✓ Correct examples: seed, freedom, mental burden, spread idea
   - ✗ Incorrect examples: breaking free from constraints, the seed of democratic thought
5. Only extract concept pairs that involve cross-domain association (i.e., the two concepts do not belong to the same conceptual domain)
6. **All output must be in English**

## Processing Steps

**Step 1 - Semantic Interpretation**: Explain the deeper meaning of this text in 1-2 sentences

**Step 2 - Association Pair Extraction**: Based on the understanding from Step 1, extract explicit concept → implicit concept association pairs

## Output Format
{{
  "interpretation": "English semantic interpretation",
  "pairs": [["explicit1", "implicit1"], ["explicit2", "implicit2"], ...]
}}
If the text contains no cross-domain associations, return an empty list [] for pairs

## Examples

**Input**: He unlocked the shackles and gained spiritual freedom

**Output**:
{{
  "interpretation": "The text uses 'shackles' to represent mental constraints. 'Unlocking' represents achieving psychological liberation.",
  "pairs": [["shackle", "mental constraint"], ["unlock", "liberate"]]
}}

**Input**: Plant the seed of democracy in his mind

**Output**:
{{
  "interpretation": "The text uses planting imagery to describe transmitting ideas. 'Seed' represents an initial idea, 'planting' represents instilling thoughts.",
  "pairs": [["seed", "idea"], ["plant", "instill"]]
}}

**Input**: Time is money

**Output**:
{{
  "interpretation": "The text equates time with money, suggesting time is a valuable resource.",
  "pairs": [["money", "valuable resource"], ["time", "asset"]]
}}

**Input**: He runs very fast

**Output**:
{{
  "interpretation": "This is a literal description without deeper associative meaning.",
  "pairs": []
}}

## Now please process the following text

**Input**: {text}

**Output**:
"""

*Figure 15.* Prompt for extracting metaphorical concept pairs.

---

**Prompt for Identifying Visual Elements**

{video}

prompt = \
"""<< Instruction >>
Watch this video carefully and extract all key content elements that appear in the video. These elements will be used for metaphor understanding analysis.

<< Requirements >>
1. Extract all significant visual elements, objects, actions, scenes, symbols, and any notable content.
2. Be comprehensive - don't miss any potentially meaningful elements.
3. Each keyword should be concise but descriptive.
4. Include both concrete objects and abstract concepts if they are clearly presented.
5. Output in JSON format.

<< Output Format >>
```json
{
"keywords": ["keyword1", "keyword2", "keyword3", ...]
}
```"""

*Figure 16.* Prompt for identifying visual elements.

---

**Prompt for Generating Video Metaphor Interpretation**

{video}
{title}

prompt = \
"""<< Instruction >>
Analyze the metaphorical logic in this video, i.e., what ideas are implicitly expressed through the content presented.

{title}

<< Requirements >>
1. Thoroughly identify all video content that contains metaphors.
2. Analyze the underlying ideas of the metaphors deeply and accurately.
3. You may refer to the external knowledge below for reference, but your analysis must be grounded in the actual video content, the reference is just for inspiration, do not rely on the reference completely.
4. Avoid baseless assumptions or forced interpretations.
5. If there are multiple elements in the video that contain metaphorical logic, list them separately, with each entry as a concise sentence.
6. Output in JSON format as a dictionary. Each analysis entry should follow the sentence structure: "The video presents *** content, implicitly expressing *** idea."

<< Here are some examples >>
{examples}

<< External knowledge for reference >>
Based on the video content, here are some relevant metaphorical associations from a knowledge graph that may help your analysis:
NOTE: these associations are just for reference, do not completely rely them.
{external_reference}

<< Output Format >>
```json
{{
"analysis_dict": {{
"analysis_1": "The video presents the *** content, implicitly expressing the *** idea",
"analysis_2": "The video presents the *** content, implicitly expressing the *** idea"
}}
}}
```"""

*Figure 17.* Prompt for generating video metaphor interpretation.

**Body Language** ⭐

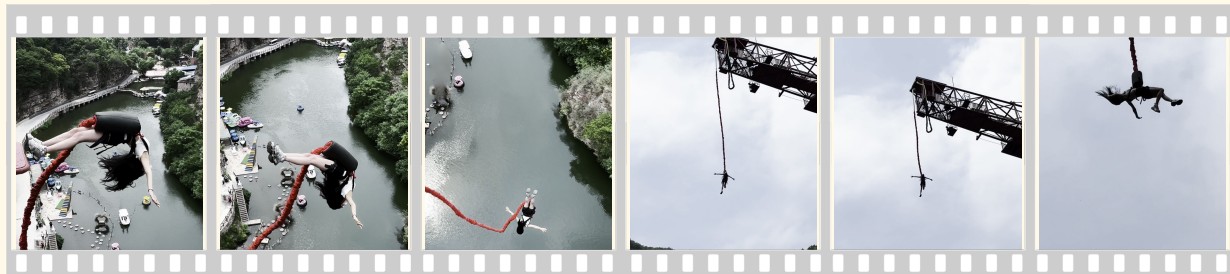

**Title**: No need title
**Metaphor Interpretation**: ["The video depicts two people walking together at a leisurely pace on a twilight street, implicitly conveying an affirmation of the belief that 'companionship is the longest confession of love' in relationships, as well as a cherishing of simple yet stable intimacy.", "The video depicts the imagery of hands holding each other with their shadows echoing on the ground, symbolizing the profound connection of 'soul resonance' and 'mutual pursuit' in love, as well as the longing for the enduring nature of intimate relationships."]

**Title**: No need title
**Metaphor Interpretation**: ["The video depicts the moment of a woman bungee jumping from a great height, implicitly symbolizing the freedom and liberation after breaking free from constraints.", "The video presents a dynamic scene of free-falling through the air, implicitly conveying the courage to face the abyss and a sense of liberation."]

*Figure 18.* Examples of Body Language. Note that most videos simultaneously contain multiple types of metaphor, we only show the dominant one in each case for convenient illustration.

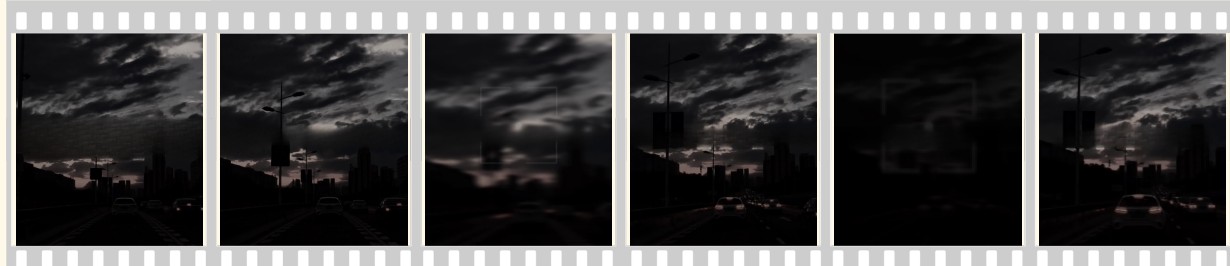

**Atmosphere Language** ⭐

**Title**: The future will be better

**Metaphor Interpretation**: ["The video depicts the dynamic fluttering of cherry blossom petals, implicitly conveying the palpitation and flow of \"love rising with the wind,\" expressing the beauty and liveliness of budding emotions.", "The video depicts a scene of pedestrians strolling amidst falling cherry blossoms, implicitly conveying an attitude of cherishing the present romance amidst the passage of time, and expressing a gentle reflection on emotions and life."]

**Title**: About my love

**Metaphor Interpretation**: ["The video depicts urban roads and moving vehicles symbolizing a 'journey,' implicitly conveying a steadfast commitment to 'companionship' and the shared anticipation of emotional bonds. This aligns with descriptions of relational states such as 'will accompany you for a long, long time' and 'he is still here,' resonating with metaphorical reflections on love and promises.", "The video depicts a dark and cloudy sky with a gloomy atmosphere, implicitly conveying the hardships in emotions and the regret of 'the thing remains but the person is no more.' It aligns with the lament over the past and the melancholy of changing relationships, supporting a symbolic interpretation of time and fate.", "The video depicts a dynamic scene of a road extending and vehicles moving forward (implying the imagery of 'moving forward'), set against a dark background with potential metaphors of hope (such as the faint light at the end of the road suggesting optimism). It subtly conveys the pursuit of 'light' in love and faith in destiny, echoing the emotional confession of 'I love you' and anticipation for the future, aligning with the logic of emotional experiences and symbolic meanings."]

*Figure 19.* Examples of Atmosphere Language. Note that most videos simultaneously contain multiple types of metaphor, we only show the dominant one in each case for convenient illustration.

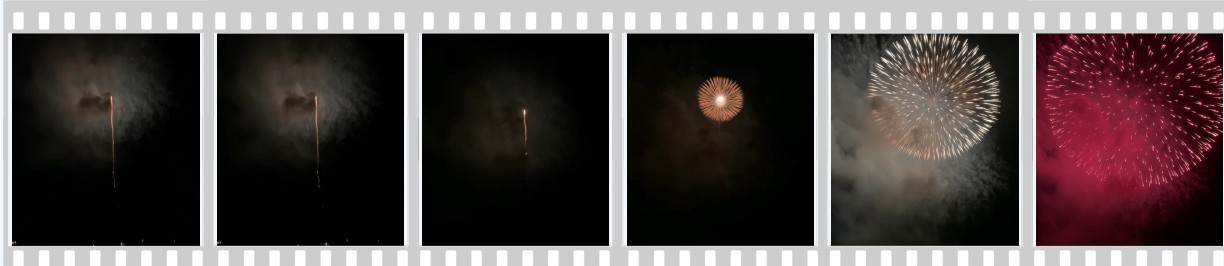

**Cultural Symbol** ⭐

**Title**: No need title
**Metaphor Interpretation**: ["The video showcases a group of Buddhist stone carvings featuring large Buddha statues, Bodhisattva statues, and numerous small Buddha figures set in a rocky environment, implicitly conveying symbolic meanings of protection and auspiciousness, as well as prayers for peace and health.", "The video shows tourists stopping and taking photos in front of the Buddha statue, implicitly expressing reverence for the religious symbol and a desire to place their wishes upon the Buddha.", "The video presents stone carvings with religious symbolism and scenes of tourists performing prayer-related actions, implicitly expressing hopes for specific wishes such as exam success, as well as psychological projections of 'divine protection' and 'receiving good luck.'"]

**Title**: Effort in my life
**Metaphor Interpretation**: ["The video depicts the dynamic process of fireworks ascending and gradually blooming in the sky, subtly conveying the idea that growth is a process of time's accumulation. Just like fireworks progress from building momentum to a dazzling display, it symbolizes life's journey of gradual accumulation over time, ultimately reaching an ideal state.", "The video depicts the visual spectacle of fireworks bursting into vibrant colors in an instant, only to fade away quickly, implicitly conveying a sense of cherishing life's beautiful moments. It also uses the fleeting bloom of fireworks as a metaphor for expressing wishes and aspirations for the future, such as 'landing safely' or 'getting into one's dream university.'"]

*Figure 20.* Examples of Cultural Symbol. Note that most videos simultaneously contain multiple types of metaphor, we only show the dominant one in each case for convenient illustration.

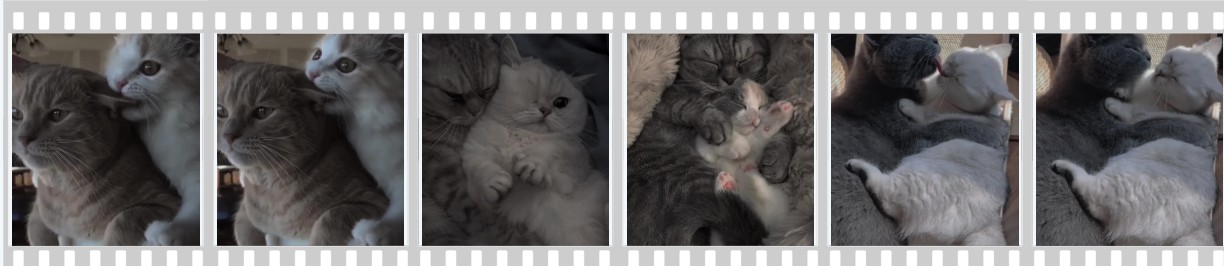

**Naturalistic Symbol** ★

**Title**: Broke up, alone

**Metaphor Interpretation**: ["The video depicts a puppy walking slowly along a slippery rural road in rainy and snowy weather, implicitly conveying feelings of longing for someone and the sorrows within emotions.", "The video depicts desolate mountains, withered grass, and an empty environment, implicitly expressing feelings of inner loneliness and helplessness, as well as a sense of letting go of past emotions.", "The video presents a somber color palette, falling snowflakes, and a desolate overall atmosphere, implicitly conveying emotional confusion defined by love and the quest for emotional belonging."]

**Title**: No need title

**Metaphor Interpretation**: ["The video depicts a tender moment of a white cat licking a gray cat's ear, implicitly conveying the warm essence of mutual care and reciprocated affection in relationships.", "The video depicts a scene of two cats snuggling and sleeping together, implicitly conveying a metaphorical projection of long-term companionship and steadfastly maintained relationships.", "The video depicts two cats grooming each other in the sunlight, implicitly conveying a beautiful interpretation of mutual dependence and peaceful companionship in intimate relationships."]

*Figure 21.* Examples of Naturalistic Symbol. Note that most videos simultaneously contain multiple types of metaphor, we only show the dominant one in each case for convenient illustration.

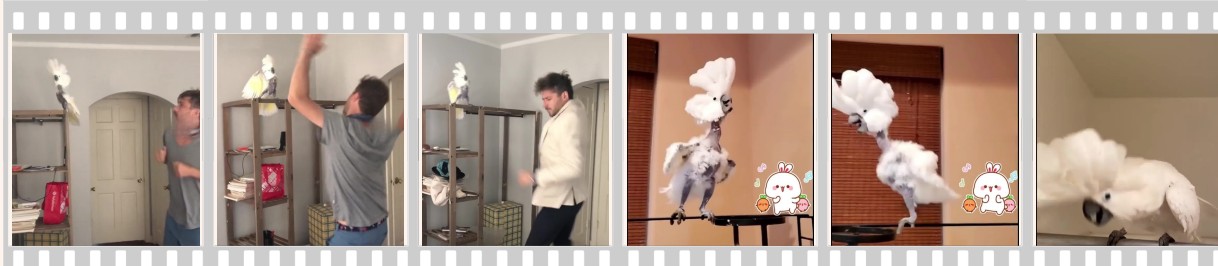

**Causal Montage** ⭐

**Title**: No need title
**Metaphor Interpretation**: ["The video depicts a scene where a boy is cleaning the blackboard while a girl secretly takes a group photo by the door, subtly conveying the metaphor of carefully preserving unspoken crushes and beautiful memories in youth.", "The video depicts a scene transition from a campus blackboard to an outdoor stroll, subtly conveying the natural growth of youthful emotions, as well as the progression of relationships from secret crushes to open affection, and from campus life to the wider society.", "The video shows a couple in wedding attire taking photos together, echoing the campus scenes at the beginning. This implicitly conveys the beautiful expectation of 'youth never parting, love lasting forever,' as well as the emotion of unrequited love finally coming to fruition."]

**Title**: No need title
**Metaphor Interpretation**: ["The video depicts the parrot's initially disheveled feathers and sluggish movements, implicitly conveying a state of inner repression and low spirits.", "The video depicts the process of a parrot gradually becoming excited and dancing under its owner's influence, implicitly conveying the positive impact of external companionship and positive guidance on psychological state.", "The video depicts the parrot in an exaggerated state of fluffed-up feathers and liberated freedom, implicitly conveying the sense of unfurling and transformation after emotional release."]

*Figure 22.* Examples of Causal Montage. Note that most videos simultaneously contain multiple types of metaphor, we only show the dominant one in each case for convenient illustration.

**Analogical Montage** ⭐

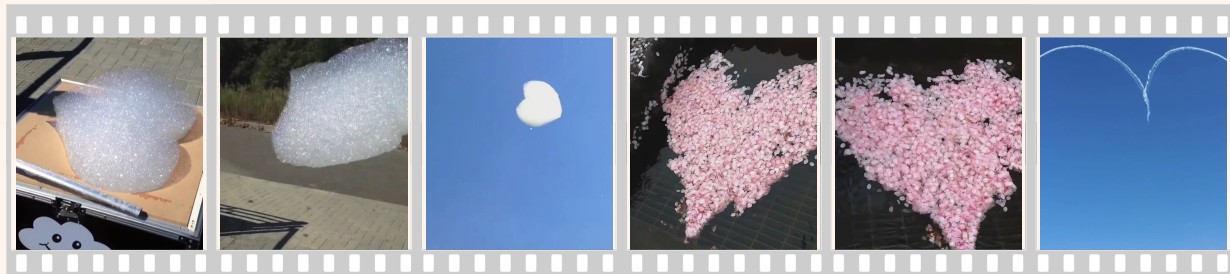

**Title**: No need title
**Metaphor Interpretation**: ["The video depicts a person chained and imprisoned in a transparent cube, implicitly expressing the self-confinement caused by appearance anxiety and the sense of oppression under societal aesthetic pressures, aligning with the personal projection of 'I used to be extremely anxious about my appearance.'", "The video depicts scenes of the self reflected in a mirror and oppressed by external appearance standards, implicitly conveying reflection on the idea that 'appearance is not the sole criterion for judgment' and resistance against monolithic beauty standards, resonating with the consensual interpretation that 'beauty should not be defined.'", "The video depicts a solitary figure floating in the universe, implicitly conveying the confusion of losing self-worth and the yearning for self-acceptance. It aligns with the in-depth exploration of the relationship between appearance and self-worth, supporting the metaphorical logic of reflecting on societal aesthetics."]

**Title**: No need title
**Metaphor Interpretation**: ["The video depicts bubbles being lifted to form a heart shape and flying away, subtly expressing a longing for the pure romance of budding love and the deep affection of 'love rising in the east and setting in the west'.", "The video depicts cherry blossom petals gradually forming a heart shape in the water, subtly conveying a longing for lasting affection and cherishing romantic moments.", "The scene in the video where an airplane leaves a heart-shaped contrail in the sky implicitly conveys the longing for long-distance love, as well as the unchanging feelings and wishes hidden in the passage of time."]

*Figure 23.* Examples of Analogical Montage. Note that most videos simultaneously contain multiple types of metaphor, we only show the dominant one in each case for convenient illustration.

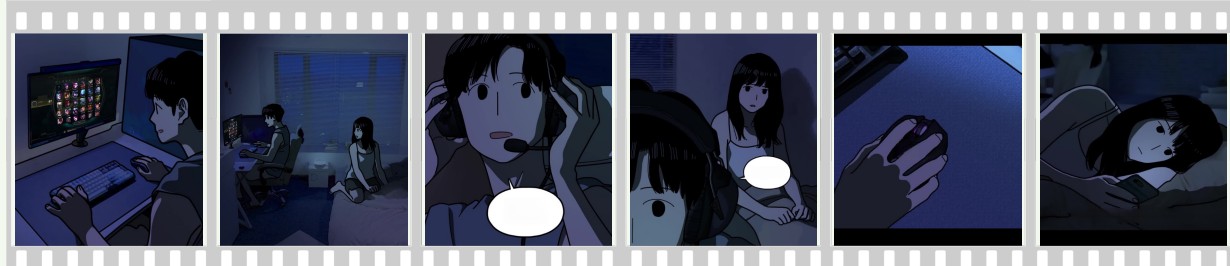

**Surreal Narrative**

**Title**: No need title

**Metaphor Interpretation**: ["The video presents a concentrated discussion in the comments section about the phenomenon of staying up late and its consequences such as kidney deficiency, etc. Commenters actively share their experiences of staying up late and relate the content to the late-night culture among young people and health warnings. This implicitly reflects the audience's resonance with and reflection on unhealthy lifestyle habits under the pressures of modern life, as well as their deep awareness and vigilance regarding the harms of staying up late.", "", "The video depicts a cartoon fox character showing visual changes such as exhaustion, obesity, oily face, and hair loss due to staying up late, with superimposed text to reinforce the effect. This implicitly symbolizes the multiple health damages caused by sleep deprivation and serves as a warning and admonition against the unhealthy lifestyles of contemporary young people who are trapped by life pressures."]

**Title**: No need title

**Metaphor Interpretation**: ["The video depicts a scene of playing games at night, implicitly conveying emotions related to 'escaping from the pressures of reality during growth'—late night, which should be a time for rest, is instead filled with gaming, metaphorically representing adults seeking temporary immersion as a mental escape when facing life's responsibilities.", "The video portrays a solemn expression during communication with a partner, implicitly conveying the idea of 'transferring responsibilities in intimate relationships.' The serious atmosphere during the interaction, rather than a lighthearted one, suggests that the romantic relationship has transitioned from mere emotional companionship to a phase of jointly confronting real-life issues. This aligns with the deeper interpretation of 'a boy becoming a man in just an instant.'", "The video presents detailed actions of shaving, implicitly conveying the idea of 'self-identification with a mature identity'—shaving is a symbolic act marking a male's transition from adolescence to adulthood. The repetitive or natural shaving motions metaphorically represent the individual's gradual acceptance of the 'adult' identity through life's trials, echoing deeper interpretations of 'maturity' and 'growing up.'"]

*Figure 24.* Examples of Surreal Narrative. Note that most videos simultaneously contain multiple types of metaphor, we only show the dominant one in each case for convenient illustration.

**Performative Narrative** ⭐

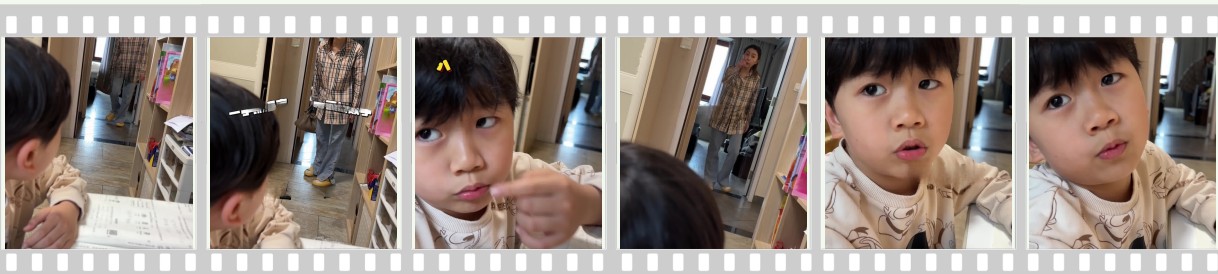

**Title**: No need title
**Metaphor Interpretation**: ["The video presents a visual scene of 'children surrounded by multiple learning tasks with no breathing space', implicitly conveying the passive situation of children under the 'leek-cutting' style of education, and transmitting the helplessness of individuals being consumed in the involution of education.", "The video presents a striking contrast between the mother's demands and the child's needs, implicitly conveying the intergenerational differences in the perception of the 'meaning of growth.' It highlights the conflict between traditional educational values and the innate needs of children, reinforcing deeper reflections on the idea that 'the meaning of life should not be solely about enduring hardship.'", "The video depicts 'a child gesturing multi-directional movements linked to the imagery of an octopus,' subtly conveying how educational pressures constrain children's natural instincts, reflecting the reality that 'holistic development' has been distorted into 'comprehensive pressure.'"]

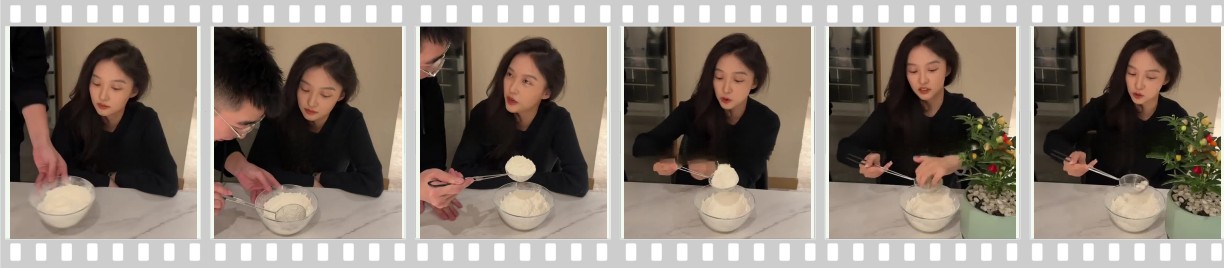

**Title**: Couples talk about emotional relationship
**Metaphor Interpretation**: ["The video depicts a scene where a man uses a slotted spoon to scoop flour, only for it to continuously leak through, subtly conveying the idea that in a single emotional relationship, feelings are prone to dissipate and difficult to stabilize.", "The video shows a woman ultimately taking the initiative to place stones on a colander, implicitly conveying an attitude of rejecting emotional PUA and maintaining clarity and independence in relationships."]

*Figure 25.* Examples of Performative Narrative. Note that most videos simultaneously contain multiple types of metaphor, we only show the dominant one in each case for convenient illustration.

