# OpenReview forum: "MetaphorVU: Towards Metaphorical Video Understanding"
_ICML.cc/2026/Conference — ICML 2026 spotlight_

### Official Review · Reviewer_uYnU · 2026-03-09

**Soundness:** 3
**Presentation:** 3
**Significance:** 2
**Originality:** 3
**Overall Recommendation:** 5
**Confidence:** 4

**Summary:**

This paper introduces MetaphorVU-Bench, a new benchmark for metaphorical video understanding built from real-world short videos and organized by an 8-way taxonomy grounded in multimodal metaphor theory. Using this benchmark, the authors show that current MLLMs lag far behind humans, largely due to failures in cross-domain mapping from visual elements to abstract concepts. To address this, they propose MetaphorBoost, an inference-time framework that augments MLLMs with a metaphor-specific knowledge graph to improve mapping, yielding consistent (though modest) gains across multiple models and accompanied by ablations.

**Compliance With Llm Reviewing Policy:**

Affirmed.

**Final Justification:**

This paper introduces MetaphorVU-Bench for the systematic evaluation of metaphor understanding in videos, alongside an enhancement framework, MetaphorBoost. I find the problem formulation well-defined, and the benchmark construction demonstrates significant systemicity.

My initial review raised concerns regarding the distinction from related works (specifically MMR-V), potential cultural biases due to the single-platform data source, and missing implementation details such as input frame counts, which hindered reproducibility. I also requested further clarification on the reliability of the LLM-Judge, error analysis, and human evaluation protocols.

The authors provided a comprehensive rebuttal that effectively addresses these points. They clarified the positioning of this work relative to MMR-V and committed to including this discussion in the final version. They also confirmed a uniform input setting of 50 frames, provided validation for LLM-Judge consistency against human ratings, and detailed their error analysis and human evaluation processes. Regarding cultural bias, the authors candidly acknowledged it as a limitation and a clear direction for future research.

Overall, the rebuttal has improved the paper's completeness and persuasiveness. Therefore, if the authors can incorporate the additional notes and experimental details from the rebuttal into the final version, I would be inclined to support the acceptance of this paper.

**Key Questions For Authors:**

1. Question type: The paper employs an LLM-Judge for evaluation, yet metaphorical understanding differs from traditional open-ended QA. Metaphors are subjective and evoke diverse human interpretations, making it difficult to establish a unique ground truth. Given the paper's own finding that models struggle to grasp human metaphors, is it scientifically sound to use a model (which itself lacks strong metaphorical capability) as a judge? Would a multiple-choice format, as used in benchmarks like II-Bench [2] and MMR-V [1], be more objective and reliable?
2. Cultural Background Analysis: Given that this is a metaphorical comprehension task, has the article undergone an analysis of cultural bias? This includes whether the video source platform is a single culture, whether the annotators come from the same cultural background, and the extent to which this affects data bias; these points should be explained.
3. Regarding Table 3, how large is the manual sample used for error analysis? What were the annotation guidelines? Were there strategies for cross-validation among annotators?
4. The details regarding the human testing in the main experiment (Table 2) are missing. What were the educational backgrounds, demographics, and cultural backgrounds of the human participants? Were they diverse and representative?

[1] Zhu K, Jin Z, Yuan H, et al. MMR-V: What's Left Unsaid? A Benchmark for Multimodal Deep Reasoning in Videos[J]. arXiv preprint arXiv:2506.04141, 2025.
[2] Liu Z, Fang F, Feng X, et al. Ii-bench: An image implication understanding benchmark for multimodal large language models[J]. Advances in Neural Information Processing Systems, 2024, 37: 46378-46480.

**Limitations:**

yes

**Strengths And Weaknesses:**

# Strength
1. This paper defines a concrete, theoretically supported taxonomy of video metaphors, enhancing the scientific basis for metaphor comprehension tasks. It comprehensively covers a variety of metaphor tasks.
2. The manuscript is well-written and easy to follow. The inclusion of data samples and example questions to helps readers understand more clearly.
3. The work is comprehensive; beyond proposing a benchmark, it provides an inference-time framework, MetaphorBoost, which offers inspiration for future research in metaphorical understanding.

# Weakness
1. Incomplete Related Work: The authors’ claim of presenting "the first comprehensive benchmark for metaphorical video understanding" is open to debate. Prior work, such as MMR-V (ICLR 2026) [1], categorizes video reasoning into explicit and implicit types. The concept of implicit reasoning in MMR-V is highly similar to the metaphorical understanding discussed in this paper, and there are also some overlaps in task definitions. The authors should discuss this.
2. The benchmark videos are exclusively sourced from Kuaishou, a short-video platform. Metaphors are inherently tied to cultural contexts. A single platform may limit the diversity and reliability of the data due to cultural factors. Incorporating videos from platforms like TikTok or YouTube could help mitigate cultural bias.
3. The paper fails to specify key implementation details, particularly the number of video frames input into the models (e.g., 16, 32, or 64 frames). This omission hinders reproducibility and makes it difficult to assess whether the comparisons between different models are fair.

---

> ### Author Rebuttal · Authors · 2026-03-31
>
> Thank you for your valuable feedback. We have carefully prepared responses to address your concerns.
>
> > W1. Related Work (MMR-V)
>
> Thank you for providing a highly valuable related work. Upon careful comparison, we would like to kindly point out that **the core unique value of our work lies in the systematicness and depth on metaphorical video understanding compared with MMR-V.**
>
> Specifically, MMR-V aims to assess a broad spectrum of reasoning abilities, where metaphor-related content appears as one of many test scenarios rather than a dedicated focus. In contrast, our work focuses specifically on metaphorical video understanding. We construct a systematic taxonomy of video metaphor and carefully curate a benchmark spanning diverse metaphor types and topics, thereby enabling more comprehensive and fine-grained analysis of MLLMs' metaphorical video understanding capability.
>
> From a broader perspective, our work and MMR-V can complement each other, jointly enabling a deep evaluation of high-order cognitive capabilities. We will discuss MMR-V in detail in the related work section in the revised paper. And we will also consider revising related statements to more precisely convey the unique positioning of our work.
>
> > W2. exclusively sourced from Kuaishou, and Q2. Cultural Background Analysis
>
> We acknowledge this limitation and kindly point out that **our goal is to establish a systematic benchmark, while the exploration of cultural variations is an important direction for future research.**
>
> Since the primary objective of this work is to construct a systematic benchmark for metaphorical video understanding, we select Kuaishou as our data source, as it provides sufficient topical diversity, a wide variety of metaphor types, and an abundance of metaphorical videos, making it well-suited for building a diverse, representative, and well-structured benchmark benchmark.
>
> Cross-platform and cross-cultural expansion are important directions for future work. Our data construction pipeline offers an efficient and scalable solution for filtering and annotating metaphorical videos, and can be readily extended to other platforms such as TikTok and YouTube. Additionally, addressing cultural bias of video metaphor also remains a key direction in future. Thank you for important suggestions and we will provide a detailed discussion of this in the limitations section of revised paper.
>
> > W3. implementation details
>
> To ensure a fair comparison, we uniformly input 50 evenly sampled frames to all MLLMs. Additionally, we will supplement the revised version with more details and fully open-source data and code to facilitate easy re-implementation.
>
> > Q1. Question type and LLM-Judge for evaluation
>
> **We ensure the reliability of the LLM Judge by providing reference answers and a well-defined scoring guideline, and validate its effectiveness through consistency with human scores.**
>
> Specifically, as detailed in Appendix C.2, rather than open-ended free-form scoring, we require the LLM Judge to assign scores based on reference answers and a procedural scoring guideline, which mitigates subjective bias at the design level. To further validate its reliability, we compare the LLM Judge scores with human scores, and as detailed in Appendix C.3, the consistency reaches a Spearman correlation of 0.85 (p < 0.001), demonstrating strong and statistically significant agreement. Therefore, under our reference-based and procedural scoring setup, the LLM Judge is capable of achieving relatively reliable scoring.
>
> Regarding question type, during preliminary experiments, we explored a multiple-choice format but found that MLLMs tended to exploit cues from the answer options rather than genuinely reasoning, consistent with findings in many previous studies[1,2]. We therefore adopted the open-ended QA format for evaluation.
>
> [1] Which of these best describes multiple choice evaluation with llms? a) forced b) flawed c) fixable d) all of the above. (Nishant et al., ACL 2025)
> [2] Does Question Really Matter? The Attribution of Answer Bias in LLM Evaluation. (Cao et al., AAAI 2026)
>
> > Q3. Regarding Table 3 (details of error analysis)
>
> We conducted Table 3 error analysis in two stages. First, we randomly sampled 100 instances and identified 4 common error types through observation and discussion. Second, 3 annotators with relevant domain expertise independently annotated all 860 instances, judging the presence or absence of each error type as four binary classification tasks. Final labels were determined by majority voting. Due to the 5,000-character limit, detailed annotation guidelines and inter-annotator agreement metrics will be reported in the revised paper.
>
> > Q4. details regarding the human testing
>
> We recruited a PhD researcher specializing in metaphor comprehension to take the test as an upper-bound reference. In the future, we will consider recruiting multiple annotators with diverse demographics and cultural backgrounds.

---

> > ### Author Rebuttal · Reviewer_uYnU · 2026-04-01
> >
> > Thanks for the authors' detailed response. Their reply largely resolved my concerns. I would like to raise my rating to 5.

---

> > > ### Author Response · Authors · 2026-04-02
> > >
> > > Thank you for reviewing our rebuttal. We are very pleased that the previous rebuttal addressed your concerns. We will carefully revise the paper in accordance with your suggestions in order to present the work at a higher quality.

---

### Official Review · Reviewer_HvpW · 2026-03-12

**Soundness:** 2
**Presentation:** 3
**Significance:** 3
**Originality:** 3
**Overall Recommendation:** 4
**Confidence:** 4

**Summary:**

The paper 1) introduces a new dataset and benchmark for visual metaphor understanding in videos; 2) evaluates API, open-weight and reasoning models; 3) designs a method to improve performance on the benchmark. The benchmark is constructed using a novel taxonomy and human validation with automatic filtering.

**Compliance With Llm Reviewing Policy:**

Affirmed.

**Final Justification:**

The authors largely resolved my questions though some concerns remained that could not be addressed in the scope of the rebuttal. The main concern is the reliance on LLM-Judge as the only evaluation method. I maintain my overall positive score.

**Key Questions For Authors:**

See weaknesses

**Limitations:**

Authors did not reflect on potential societal impact of training models to understand metaphors in video media content (e.g., censorship, surveillance).

**Strengths And Weaknesses:**

***Strengths:***

- Detailed new taxonomy of visual metaphors in videos based on prior theoretical work on multimodal metaphor understanding.

- Rigorous data collection pipeline, though some details on human validation were lacking.

- Results offer interesting insights, for example that inference-time scaling and post-training methods do not yield much improvement on the visual metaphor interpretation task.

- New method is introduced based on knowledge graph augmentation, appearing to yield some improvements.

***Weaknesses:***

- **Video Metaphor Taxonomy Justification:** (minor weakness) while the taxonomy is very detailed and appears to have theoretical basis, it combines a wide range of theories without proper justification why those theories were selected. Some justifications also did not seem to fully align, like the Body Language category in the Film Mise-en-scene theory; others seemed incomplete, like the symbolism theory only including cultural or naturalistic symbols. Overall, the justification seemed rather post-hoc rather than derived ground-up from theory, as claimed.

- **Missing details on human annotation:** The benchmark's final construction step is manual annotation, but the details on it are not reported, only that it was done by a "human team". How many annotators were involved? What were their qualifications? What are the statistics on how the 2 reviewers changed the initial annotations in the interpretation annotation stage? It is also unclear why for human baseline results in Table 2, additional 100 annotations were collected?

- **Missing details on LLM-Judge verification:** The biggest weakness of the paper is lack of a verifiable task to concretely measure model performance without the confounds of LLM judge (unlike, e.g., previous work that has proposed framing metaphor understanding as a binary detection [1] or entailment [2] tasks), and the scarce details on LLM-judge validation. Since all evaluation is done with LLM Judge, judge verification is crucial for the paper. Paper mentions "human annotators" scoring the items but very few details are provided. How many annotators? Was the human score determined by an average? Were there any instances of large disagreement between llm judge and human scores? Did llm judge exhibit bias towards some language model output that humans did not? What measure were taken to mitigate known LLM judge biases?

- **Lack of statistical robustness of the results:** Main results present raw scores without any kind of confidence intervals around them. Given the marginal improvement of the proposed MetaphorBoost method (averaging 2.3 points for Gemini), it is unclear whether this difference is at least statistically significant. Authors should consider adding at least bootstrap 95% confidence intervals.

- **Missing error analysis details and justification for mapping improvement:**
    - How was the error analysis conducted? how many samples analyzed? how many human raters? How were proportions in Table 3 computed?
    - The claim that variation in metaphor type performance results support the need for improved cross-domain mapping appears unsupported. Why do those types contain "richer metaphorical visual elements" compared to for example natural/cultural symbolism or atmospheric language? The latter seem more metaphoric than e.g. body movement.

- **Missing details on method analysis:** The claim on the decline of mapping errors as a result of the method is not sufficiently substantiated. Virtually no details are provided on how this analysis was conducted.

- **Missing related work on (visual) metaphor understanding:**
    - [1] [Looking Beyond the Pixels: Evaluating Visual Metaphor Understanding in VLMs](https://aclanthology.org/2025.findings-emnlp.1257/) (Kundu et al., Findings 2025)
    - [2] [Understanding Figurative Meaning through Explainable Visual Entailment](https://aclanthology.org/2025.naacl-long.1/) (Saakyan et al., NAACL 2025)
   - Missing vast amount of methodological related work on inference-time improvement of LLMs with knowledge graphs or knowledge graph construction (e.g. [3] [COMET: Commonsense Transformers for Automatic Knowledge Graph Construction](https://aclanthology.org/P19-1470/) (Bosselut et al., ACL 2019)), metaphor understanding with knowledge graphs ([4] [MERMAID: Metaphor Generation with Symbolism and Discriminative Decoding](https://aclanthology.org/2021.naacl-main.336/) (Chakrabarty et al., NAACL 2021)), metaphor understanding datasets ([5] [FLUTE: Figurative Language Understanding through Textual Explanations](https://aclanthology.org/2022.emnlp-main.481/) (Chakrabarty et al., EMNLP 2022)).

Nitpicks:
- line 419: agumentatio -> agumentation
- Unclear dataset selection for metaphor graph construction. Why were these datasets chosen? Why not use published datasets, e.g. [5]?

---

> ### Author Rebuttal · Authors · 2026-03-31
>
> Thank you for your valuable feedback. We have carefully prepared responses to address your concerns.
>
> > Video Metaphor Taxonomy Justification
>
> **Since no systematic taxonomy for video metaphor exists, constructing one is inherently exploratory. We combine theoretical investigation with empirical observation to ensure the video metaphor taxonomy is both grounded and comprehensive.**
>
> Specifically, we first survey relevant literature (see Appendix A) to identify candidate metaphor types as an initial framework. We then iteratively refine these metaphor types through extensive observation of real-world video platforms, arriving at 8 final categories, which are used to support our benchmark for fine-grained analysis.
>
> > details on human annotation
>
> We will address the following two concerns respectively: the details of manual annotation and the setup of the human baseline in Table 2.
>
> Regarding manual annotation:
>
> (1) As described in Line 239–244, the human team responsible for annotation consists of 1 annotator for writing the initial interpretation, and the other 2 annotators for reviewing.
>
> (2) To ensure annotation quality, we recruited 3 professional annotators with Master's degrees in literature and extensive experience in open-ended text annotation tasks.
>
> (3) Among the 860 annotated instances, 803 initial interpretations were directly approved by both reviewers, 42 were rejected by at least one reviewer, and 15 were rejected by both. The disputed cases underwent iterative discussion and refinement among the 3 annotators until consensus was reached.
>
> Regarding the human baseline in Table 2:
>
> The human baseline serves as an upper bound reference for model performance. To ensure it truly reflects optimal human capability, we recruited a PhD researcher specializing in metaphor comprehension to take the test. Due to the high cost, we randomly sampled 100 data from the full 860 data for baseline construction.
>
>
> > details on LLM-Judge verification
>
> **We ensure the reliability of LLM Judge by providing reference answers and establishing a well-defined procedural scoring process, and we validate its effectiveness by demonstrating consistency with human scores.**
>
> First, as detailed in Appendix C.2, rather than allowing the LLM to perform open-ended free-form scoring, we required it to assign scores based on reference answers and a well-defined hierarchical scoring guideline, which specifies a detailed analytical procedure for judging and provides explicit criteria for each score level. This reference based and procedural scoring method mitigates the risk of subjective bias of LLM Judge at the design level, thereby ensuring the reliability of the evaluation results to a reasonable extent.
>
> To further validate the reliability of LLM Judge, we recruited 3 annotators to conduct independent human scoring, and aggregated their scores using a median strategy to obtain the final human scores. As detailed in Appendix C.3, the statistical results show that the consistency between human scores and LLM Judge scores reaches 0.85, with a corresponding p-value of 3e-20 (p < 0.001), which is highly significant and sufficiently demonstrates the reliability of the LLM Judge in our task.
>
> > error analysis details and justification for mapping improvement
>
> Regarding error analysis details:
>
> We conducted error analysis in two stages to ensure both coverage and rigor. First, we randomly sampled 100 instances from outputs of Gemini-3-Pro and Qwen3-VL-8B-Thinking, and identified 4 common error types through observation and discussion. Second, 3 annotators with relevant domain expertise independently annotated all 860 instances, judging the presence or absence of each error type as four binary classification tasks. Final labels were determined by majority voting, from which we computed the proportions in Table 3. Detailed annotation guidelines and inter-annotator agreement metrics will be reported in the revised paper.
>
> Regarding justification for mapping improvement:
>
> Based on our observations, the latter four types generally contain multiple scenes or storylines. While there are some cases with frequent scene changes in the first four types as well, this is not a common phenomenon. Therefore, we take the performance variation across metaphor types as secondary and supplementary evidence of improved cross-domain mapping.
>
> > details on method analysis
>
> Regarding the decline of mapping errors demonstrated in Figure 7, we adopt the same analytical approach as applied to Table 3, as explained above. We will provide a more detailed elaboration on this in the revised paper.
>
> > statistical robustness, related work, nitpicks, potential societal impact, and more details of above response
>
> Due to the 5,000-character limit, we appreciate these insightful suggestions and will provide more detailed elaboration in the revised paper. In particular, regarding the related works you provided, we will cite all of them and discuss each one in detail.

---

> > ### Author Rebuttal · Reviewer_HvpW · 2026-04-01
> >
> > Thank you my concerns are partially resolved. I am not fully convinced that LLM-as-a-Judge provides unbiased evaluation.  The task relies on a lot of reasoning and interpretation when comparing to reference. The authors did not respond to these questions: Were there any instances of large disagreement between llm judge and human scores? Did llm judge exhibit bias towards some language model output that humans did not? What measure were taken to mitigate known LLM judge biases?
> >
> > Concerns that I think are not possible to address within the rebuttal period: only 1 human used for human performance. the human performance is only an estimate on 100 instances. it does not reflect performance of lay humans which may be behind the LLM (so perhaps need to reframe to "expert human performance").
> >
> > The evidence for the cross-domain mapping is still unclear to me.
> >
> > For statistical robustness, given the small sample of 860 items, it is quite important to support the main claim of method's improvement over the baseline. (e.g., bootstrap confidence intervals and p-value).

---

> > > ### Author Response · Authors · 2026-04-02
> > >
> > > Thank you for reviewing our rebuttal. We are very pleased that the previous rebuttal addressed some of your concerns. In the following, we have carefully prepared further detailed explanations for some issues that remain unresolved.
> > >
> > > > statistical robustness
> > >
> > > Below are our statistical robustness results, including bootstrap 95% confidence intervals and p-values, which demonstrate that our results are statistically robust.
> > >
> > >
> > > |                  | MetaphorBoost (Gemini-3-Pro) | MetaphorBoost (Qwen2.5-VL-7B-Instruct) | MetaphorBoost (Qwen3-VL-8B-Thinking) |
> > > |------------------|------------------------------|----------------------------------------|--------------------------------------|
> > > | bootstrap 95% CI | [0.05, 0.41]                 | [0.28, 0.58]                           | [0.22, 0.55]                         |
> > > | p-value          | 0.0044                       | 0.0001                                 | 0.0001                               |
> > >
> > > > LLM-as-a-Judge, human performance, evidence for the cross-domain mapping
> > >
> > > These issues may not be fully addressed during the rebuttal stage. We will carefully consider your suggestions and provide more detailed clarifications on these aspects in the revised manuscript, in order to present the work at a higher quality.

---

### Official Review · Reviewer_bbrD · 2026-03-12

**Soundness:** 3
**Presentation:** 4
**Significance:** 3
**Originality:** 3
**Overall Recommendation:** 5
**Confidence:** 4

**Summary:**

This paper introduces MetaphorVU-Bench, a benchmark for metaphorical video understanding. It’s built upon 8 metaphoric categories and curated from real-world data through a well-structure pipeline. The paper reports empirical results on a range of models based on the benchmark. Through detailed analysis, it found that the majority of failures do not stem from recognition error, but rather from models’ inability to connect visual elements to underlying concepts. To address this problem, the paper further introduces MetaphorBoost that utilizes a metaphorical knowledge graph as external cognitive scaffold to augment visual entity to concept mapping and demonstrates improved performance.

**Compliance With Llm Reviewing Policy:**

Affirmed.

**Final Justification:**

fully addressed my questions. I will keep my score.

**Key Questions For Authors:**

-	Could you elaborate “linking visual elements to underlying concepts is the key to improving MLLMs performance”, what “underlying concepts” are you referring to? Metaphorical concepts? Or semantic concepts e.g., through visual grounding?
-	In Figure 7, which base model is compared against?

**Limitations:**

No limitation is explicitly discussed.

**Strengths And Weaknesses:**

Strengths:
- The paper is well motivated, addressing an under-explored area on metaphorical video understanding.
- The benchmark generation pipeline is well described ensuring data quality
- It made available of a metaphor-specific knowledge graph, first of its kind, to facilitate research in the general area of metaphor understanding.
- Proposes a technical which uses metaphorical knowledge-graph to better connect visual elements and underlying concepts and establishes a strong baseline for future research.

Weaknesses:
- Although it’s not exactly the same, I’m wondering whether some previous work on sarcasm would be relevant here.  It may deserve some discussion here.
- No limitation of this work is discussed.

---

> ### Author Rebuttal · Authors · 2026-03-31
>
> Thank you for your valuable feedback. We have carefully prepared responses to address your concerns.
>
> > Although it’s not exactly the same, I’m wondering whether some previous work on sarcasm would be relevant here. It may deserve some discussion here.
>
> We thank the reviewer for this important suggestion. Multimodal sarcasm research is relevant to our work and deserves discussion. **In general, multimodal sarcasm understanding and metaphorical video understanding differ in their core capability requirements and the types of implicit meanings they encompass.**
>
> In terms of core capability requirements, sarcasm primarily relies on identifying apparent contradictions among elements[1,2], whereas metaphorical video understanding requires models to perform cross-domain mapping, i.e., linking visual elements to underlying concepts.
>
> In terms of implicit meanings, sarcasm mainly focuses on conveying critical and negative thoughts[3,4], whereas metaphorical video understanding covers a broader and more diverse range of implicit meanings, as shown in Figure 4, encompassing various forms prevalent in everyday life.
>
> We will add a dedicated discussion of sarcasm-related work in the related work section of our revised paper to help readers better understand the connections and differences between the two.
>
> [1] Zhuang, Xingjie, Fengling Zhou, and Zhixin Li. "Multi-modal sarcasm detection via knowledge-aware focused graph convolutional networks." ACM Transactions on Multimedia Computing, Communications and Applications 21.5 (2025): 1-22.
> [2] Ou, Lisong, and Zhixin Li. "Multi-modal sarcasm detection on social media via multi-granularity information fusion." ACM Transactions on Multimedia Computing, Communications and Applications 21.3 (2025): 1-23.
> [3] Wang, Peng, et al. "S3 agent: Unlocking the power of VLLM for zero-shot multi-modal sarcasm detection." ACM Transactions on Multimedia Computing, Communications and Applications 21.11 (2025): 1-16.
> [4] Wang, Xinyu, Yue Zhang, and Liqiang Jing. "Can Large Vision-Language Models Understand Multimodal Sarcasm?." Proceedings of the 34th ACM International Conference on Information and Knowledge Management. 2025.
>
> > Could you elaborate “linking visual elements to underlying concepts is the key to improving MLLMs performance”, what “underlying concepts” are you referring to? Metaphorical concepts? Or semantic concepts e.g., through visual grounding?
>
> The 'underlying concepts' referred to in the paper are 'metaphorical concepts'. For example, as illustrated in Figure 1, the visual elements depicted are tailcoat-wearing pigs, and the underlying concept (i.e. metaphorical concept) they represent is the ruling class. Thank you for your helpful comment, we will provide a clearer explanation of this term in the revised manuscript.
>
> > In Figure 7, which base model is compared against?
>
> As indicated in the captions of the two sub-figures in Figure 7, our experiments incorporate both the high-performance closed-source Gemini-3-Pro and the widely-used open-source Qwen3-VL-8B-Thinking as base model, thereby ensuring the reliability of experimental conclusions. The results demonstrate that our method achieves consistent and notable performance improvements across both base models, confirming that it can deliver reliable and generalizable enhancement effects.
>
> > No limitation of this work is discussed.
>
> We appreciate the reviewer's important reminder. As the call for papers did not explicitly require a limitations section, our original submission did not include one. The following is a detailed discussion of the limitations of our paper, which we will incorporate into the revised manuscript.
>
> First, given the remarkably rapid iteration of MLLMs, it is not feasible for us to continuously track and evaluate all state-of-the-art MLLMs throughout the course of this work. As a result, several recently released MLLMs (e.g., Gemini-3.1-Pro, Qwen-3.5, and GPT-5.4) are not included in our comparisons in Table 2. To address this limitation, we plan to build and maintain a public leaderboard that supports community-contributed evaluation submissions, enabling continuous tracking and reporting of frontier model performance on metaphorical video understanding.
>
> Second, since the primary objective of this work is to construct a systematic benchmark for metaphorical video understanding sourced from real-world video platforms, we select Kuaishou as the sole data source because it offers a sufficiently rich diversity of topics and an abundance of metaphorical videos. Nevertheless, we acknowledge that relying on a single platform may introduce potential biases in cultural and stylistic coverage. In future work, we plan to re-use our metaphorical video collection and annotation pipeline to additional platforms such as YouTube and TikTok, enabling broader investigation into cross-platform and cross-cultural metaphorical video understanding.

---

> > ### Author Rebuttal · Reviewer_bbrD · 2026-03-31
> >
> > I'd like to thank the authors for addressing my questions. They are fully resolved and I'd like to maintain my high rating.

---

> > > ### Author Response · Authors · 2026-04-02
> > >
> > > Thank you for reviewing our rebuttal. We are very pleased that the previous rebuttal addressed your concerns. We will carefully revise the paper in accordance with your suggestions in order to present the work at a higher quality.

---

### Official Review · Reviewer_tSSn · 2026-03-14

**Soundness:** 2
**Presentation:** 3
**Significance:** 2
**Originality:** 3
**Overall Recommendation:** 5
**Confidence:** 4

**Summary:**

The paper proposes a benchmark, MetaphorVU, for studying how MLLMs understand videos that convey information through metaphors. 8 types of metaphors (Fig 2) are considered in the benchmark; a subset of videos are carefully curated through a multi-stage AI + human filtering pipeline; and human-authored interpretations are the main prediction target. The evaluation is performed with LLM as a judge, but shows high correlation with human judgements. Beyond evaluating many MLLMs in a zero-shot manner, the paper also identifies cross-domain mapping (between visual objects to abstract concepts) as a key limitation. It proposes MetaphorBoost that uses public text datasets to learn a knowledge graph for this mapping and enhances inference time prediction by picking concepts from the graph.

**Compliance With Llm Reviewing Policy:**

Affirmed.

**Final Justification:**

The rebuttal has addressed my main concerns. W2 and W4 have also been addressed sufficiently. I am happy to increase my rating from **borderline** (described in text) to **accept (5)**. Authors are strongly encouraged to include the relevant discussion in the main paper.

**Key Questions For Authors:**

I am **borderline** about this paper while leaning towards acceptance given the timeliness of this work on moving towards understanding of abstract concepts (metaphors). That said, I will keep my positive rating assuming that the response will address the points raised in the weaknesses.

In particular, questions raised in W1-W3 should be addressed in detail as they directly pertain to the quality of the benchmark and the need for the taxonomy. Some points of W4 and W5 may be answered. I think W5e in particular may be a simple experiment to try for at least one of the models.

**Limitations:**

Limitations are not discussed. One aspect is to do with the video dataset mostly coming from Asia (platform Kuaishou) may lead to certain cultural specificities that are not common in other parts of the world. Impact statement is ok.

**Strengths And Weaknesses:**

**Some strengths:**

S1. The dataset collection and filtration, and overall manual annotation process seems to be quite strong. Examples of the dataset are very interesting and likely challenging. As MLLMs get better at generic video understanding, abstract concepts and metaphors in particular are a good topic to work on.

S2. The MetaphorBoost idea is simple and quite neat. The knowledge graph built in such a way will be quite interesting. The performance improvements are also reasonable given the complexity of the task and the LLM judge evaluation.

S3. It is good to see a thorough evaluation and discussion of various MLLMs from closed-source to open-source to reasoning/thinking models. More importantly, I appreciate the error analysis (Sec 4.3, Table 3) is very useful to understand the current state of models.

**Some weaknesses:**

W1. P2-C2-L90-94 dismisses previous work on metaphor understanding claiming that it is on advertisement videos. This seems a little confusing to me; why is analyzing ads not enough? In fact, one could argue that ads may make the metaphors most clear, perhaps even more than some examples presented in this work (see next point). A deeper explanation of this is highly desirable. (Soundness)

W2. Fig 18-25 showcase many qualitative examples. This is great. However, I feel like some of these videos are forced to take on a metaphorical meaning instead of being observed for their actual depiction. For example, the bungee jump in Fig 18 or fireworks in Fig 20; unless this video clip is presented with some other scenario, it is strange to associate such deep meanings of liberation or beautiful life moments. (Soundness of dataset)

W3. The defined taxonomy is a bit confusing. From Appendix A, it seems like different theories are put together to identify 4 sets of binary labels. But the taxonomy is presented as having 8 classes. Furthermore, the taxonomy is not used as a target label, but primarily to report results across different classes. Therefore it is unclear why the taxonomy is presented in the first place. Additionally, Fig 2 indicates that most videos contain multiple labels, more details about this would be helpful to properly understand the benchmark. (Significance)

W4. I felt that some of the videos may not really require understanding the action / motion as much. For example, for the Kongming lantern in Fig 2, a single frame may be enough to suggest all the hopes of a brighter future. Thus, the benchmark may not be evaluating for deep video understanding.

W5. Minor comments:

a. P4-C1-L213-216: I'm not sure if filtering to videos with more comments indicates that the video is complicated and requires explanation. There are many videos (at least on Youtube) with congratulatory comments or just praising the content. This may introduce some weird confounders in the benchmark creation process.

b. P2-C1-L74: real-world video platform seems a bit abstract.

c. P5-C1-L247: Dialog / audio may be an important modality and is being removed in this work.

d. More details for the MetaphorBoost method would be desirable. It is not at a point where someone can re-implement it easily.

e. Do the models really require the video title? How would performance change without the title?

---

> ### Author Rebuttal · Authors · 2026-03-31
>
> Thank you for your valuable feedback. We have carefully prepared responses to address your concerns.
>
> > W1. why is analyzing ads not enough
>
> **Advertisement videos fail to reflect the diversity and real-world coverage of video metaphor, making them insufficient for a comprehensive evaluation of metaphorical video understanding.**
>
> First, metaphor in commercial advertisements typically revolves around product features, resulting in relatively fixed metaphorical intent and expression patterns. In contrast, our benchmark encompasses various metaphor types in Fig 2 and a wide range of topics in Fig 4, thus better reflecting the complexity of video metaphor.
>
> Furthermore, in real-world tasks such as public opinion analysis and misinformation detection, metaphorical videos predominantly originate from everyday social media rather than advertisements. Therefore, our benchmark can more faithfully reflect MLLM performance across various practical applications.
>
> > W2. feel like some of these videos are forced to take on a metaphorical meaning
>
> We would like to kindly clarify that our filtration process includes rigorous cross-validation quality control (P5-C1-L234-248) to ensure each video has clear metaphorical logic. Some less representative examples are primarily due to inadequate presentation of contextual information.
>
> First, for videos that require background for interpretation, we provide the necessary context through titles. For example, for Fig 20 (fireworks), the video title is "Effort in my life" in P28-C1-L1523, with which the fireworks can naturally be interpreted as a metaphorical expression of life's beauty and aspirations.
>
> Furthermore, since only partial frames are presented in the paper, the complete information of some videos cannot be fully conveyed. For example, for Fig 18 (bungee jump), the original video begins with a brief scene of busy working, followed by  bungee jumping, together constituting a metaphorical expression of constraints and liberation.
>
> > W3. defined taxonomy is a bit confusing
>
> We will alleviate the confusion from 3 aspects: why the taxonomy is presented, how the 8-class taxonomy is constructed, and the issue about multiple labels.
>
> First. **Without a systematic taxonomy, fine-grained analysis of MLLMs' deficiencies would be difficult to conduct. We therefore first construct such a taxonomy to facilitate result interpretation and enable targeted analysis.** For instance, in P7-C1-L341-351, MLLMs perform significantly worse on the latter four types, revealing a deficiency in integrating dispersed metaphorical elements. This taxonomy can also serve as a reusable framework for future video metaphor research.
>
> Second. To the best of our knowledge, a systematic taxonomy of video metaphor remains absent. To accomplish this pioneering task, we combine theoretical investigation with empirical observation. We first identify candidate types by referencing relevant literature (Appendix A), then refine them through extensive observation of metaphor-logic videos on real platforms, yielding 8 types. The "4 sets of binary labels" in Appendix A is simply for presentational clarity.
>
> Furthermore. Given the rich elements of videos, a video may naturally involve multiple metaphor types., And we annotate all applicable types while also identifying the most representative one during annotation. To maintain research focus and avoid getting stuck in nitpicks, we retain only the most representative type per video in Fig 2 and experiments.
>
> > W4. a single frame may be enough
>
> **Compared to complete videos, relying solely on a single frame has two key limitations: dynamic-process invisibility and interpretive ambiguity.** First, videos can convey metaphorical meanings through dynamic processes, for example, a Kongming lantern continuously drifting higher conveys "hopes soaring towards a brighter future," which a single frame can not fully express. More importantly, incomplete information in a single frame can lead to ambiguity: the same frame could depict a lantern drifting into the distance or gradually falling, carrying opposite metaphorical meanings, whereas a complete video resolves such ambiguity.
>
> > W5a. not sure if videos with more comments indicates that the video is complicated
>
> Filtering by amount of comments is just the first and coarse-grained stage, non-metaphorical videos are subsequently removed through three additional stages in Fig 3, ensuring the quality of the final dataset.
>
> > W5e. How would performance change without the title
>
> Since some video titles provide necessary context, removing them leads to obvious performance drop, overall scores are shown below.
> | | GPT-5 | Gemini-2.5-Pro | GLM-4.5V |
> |-|-|-|-|
> | w/ title | 63.7 | 61.8 | 56.8 |
> | w/o title | 56.7 | 56.3 | 52.1 |
> > W5b, W5c, W5d, Limitation, and more details of above response
>
> Due to the 5,000-character limit, we appreciate these insightful suggestions and will provide detailed elaboration in the revised paper.

---

> > ### Author Rebuttal · Reviewer_tSSn · 2026-04-01
> >
> > Thanks to the authors for their detailed response.
> >
> > W1. I understand the response. It will be good to include this discussion in the paper to reflect challenges rather than simply dismissing advertisement videos. This is resolved.
> >
> > W2, W5e. Clubbing together these 2 as they are related. Quantitatively, the title seems quite important with big improvements. Qualitatively too, it seems like the title plays a big role in identifying the metaphor. For example, the fireworks video with a negative title such as "Everyone else seems to be celebrating" could completely change the interpretation. I think this point needs to be clarified in the final paper and feels a bit like an oversell to me. At this point, the paper is "video + title" metaphor interpretation. Would like to discuss this with other reviewers / AC too.
> >
> > W3. Ok.
> >
> > W4. Given the size of the dataset, I would not expect there to be multiple kongmin lanterns drifting / floating / soaring, etc. This is what worries me, the model is possibly biased towards thinking that all lanterns must soar because the internet data is limited. I request the authors to include an experiment that shows results with a single middle frame of the video. This would help address the concern.
> >
> > W5a. I'm not worried about the precision. The recall (selection of metaphorical video) was a bit unknown. Anyway, this is a minor point and I'd be happy to consider this as resolved.
> >
> > Limitations (W6). Cultural bias due to platform may be relevant to discuss too. But I see it discussed for reviewer `uYnU`.
> >
> > ----
> >
> > In summary, it may be good if the authors can respond to further points about W2 and W4. Thank you.

---

> > > ### Author Response · Authors · 2026-04-02
> > >
> > > Thank you for reviewing our rebuttal. We are very pleased that the previous rebuttal addressed some of your concerns. In the following, we have carefully prepared further detailed explanations for the issues that remain unresolved.
> > >
> > > > W2. the paper is "video + title" metaphor interpretation
> > >
> > > As described in the paper (P5-C1-L251-255), the input to MLLMs during evaluation consists of video and title, where the title serves to provide necessary contextual information for metaphor comprehension. Thank you for your suggestion, we will provide a more detailed explanation of this setting when revising the paper.
> > >
> > > > W4. I request the authors to include an experiment that shows results with a single middle frame of the video.
> > >
> > > **Due to the dynamic-process invisibility and interpretive ambiguity, a single frame is insufficient for evaluating MLLMs' metaphorical video understanding capabilities.**
> > >
> > > The experimental results are shown below. We tested two settings: taking a single first frame and taking the middle frame across all three models, performance consistently declined, particularly on the last four task categories, where performance even dropped drastically by as much as 38 points (Gemini-2.5-Pro on the Causal M. subtask).
> > >
> > > This demonstrates that a single frame falls far short of enabling complex metaphor comprehension in the way that a complete video does, further supporting the necessity of our benchmark for evaluating the metaphorical video understanding abilities of MLLMs.
> > >
> > > |                          | Body L. | Atmosph. L. | Cultural S. | Natural. S. | Causal M. | Analog. M. | Surreal N. | Perform. N. | Average |
> > > |--------------------------|---------|-------------|-------------|-------------|-----------|------------|------------|-------------|---------|
> > > | GPT-5                    |         |             |             |             |           |            |            |             |         |
> > > | complete video           | 69.9    | 76.3        | 77.4        | 66.6        | 45.0      | 55.4       | 54.9       | 46.1        | 63.7    |
> > > | only the first frame     | 38.0    | 66.4        | 50.4        | 58.2        | 7.9       | 15.8       | 29.9       | 9.2         | 36.6    |
> > > | only single middle frame | 40.0    | 71.7        | 57.7        | 61.3        | 11.7      | 16.3       | 35.3       | 9.7         | 40.1    |
> > > | Gemini-2.5-Pro           |         |             |             |             |           |            |            |             |         |
> > > | complete video           | 65.5    | 71.3        | 74.3        | 64.4        | 53.5      | 55.7       | 52.1       | 46.9        | 61.8    |
> > > | only the first frame     | 42.8    | 64.4        | 53.5        | 50.3        | 15.0      | 21.9       | 30.9       | 21.1        | 39.5    |
> > > | only single middle frame | 41.3    | 69.1        | 55.0        | 56.3        | 15.7      | 23.6       | 37.4       | 13.2        | 41.4    |
> > > | GLM-4.5V                 |         |             |             |             |           |            |            |             |         |
> > > | complete video           | 62.7    | 67.9        | 71.9        | 62.1        | 37.6      | 50.1       | 46.1       | 38.4        | 56.8    |
> > > | only the first frame     | 40.7    | 60.1        | 48.9        | 50.8        | 18.7      | 21.3       | 30.6       | 16.6        | 37.6    |
> > > | only single middle frame | 40.2    | 65.7        | 56.7        | 55.3        | 14.2      | 21.4       | 34.4       | 20.0        | 40.4    |

---

### Decision · Program_Chairs · 2026-04-30

**Decision:**

Accept (spotlight)

**Comment:**

This paper introduces MetaphorVU-Bench and the MetaphorBoost framework to systematically test and enhance MLLMs on high-order cognitive metaphor tasks. Reviewers acknowledged the paper for its rigorous manual annotation, solid taxonomy, and the simplicity of the knowledge-graph enhancement. The authors successfully clarified points regarding dataset biases, implementation details, and LLM-as-a-judge reliability during the rebuttal. The resulting scores (5, 5, 4, 5) reflect a unified consensus on the paper contribution to the field, and the AC recommends strong accept.